# Predicting knee osteoarthritis progression using neural network with longitudinal MRI radiomics, and biochemical biomarkers: A modeling study

Ting Wang[1,2☯], Hao Liu[1☯], Wenbo Zhao[1], Peihua Cao[3], Jia Li[4], Tianyu Chen[5], Guangfeng Ruan[6], Yan Zhang[3], Xiaoshuai Wang[3], Qin Dang[3,7], Mengdi Zhang[3,7], Alexander Tack[8], David Hunter[3,9], Changhai Ding[3,10]*, Shengfa Li[1]*

1 Department of Orthopaedics, The Third People's Hospital of Chengdu, Affiliated Hospital of Southwest Jiaotong University & The Second Affiliated Hospital of Chengdu, Chongqing Medical University, Chengdu, Sichuan, China, 2 Medical Research Center, The Third People's Hospital of Chengdu, Affiliated Hospital of Southwest Jiaotong University & The Second Affiliated Hospital of Chengdu, Chongqing Medical University, Chengdu, Sichuan, China, 3 Clinical Research Centre, Zhujiang Hospital, Southern Medical University, Guangzhou, Guangdong, China, 4 Division of Orthopaedic Surgery, Department of Orthopaedics, Nanfang Hospital, Southern Medical University, Guangzhou, Guangdong, China, 5 Department of Orthopaedics, The Third Affiliated Hospital of Southern Medical University, Guangzhou, Guangdong, China, 6 Clinical Research Centre, Guangzhou First People's Hospital, School of Medicine, South China University of Technology, Guangdong, Guangzhou, China 7 Department of Orthopaedics, Zhujiang Hospital, Southern Medical University, Guangzhou, Guangdong, China, 8 Zuse Institute Berlin, Berlin, Germany, 9 Department of Rheumatology, Royal North Shore Hospital and Sydney Musculoskeletal Health, Kolling Institute, University of Sydney, Sydney, New South Wales, Australia, 10 Menzies Institute for Medical Research, University of Tasmania, Hobart, Tasmania, Australia

☯ These authors were dual first authors and contributed equally to this work.
* lisheng-fa@qq.com (SL); changhai.ding@utas.edu.au (CD)

## Abstract

### Background

Knee osteoarthritis (KOA) worsens both structurally and symptomatically, yet no model predicts KOA progression using Magnetic Resonance Image (MRI) radiomics and biomarkers. This study aimed to develop and test the longitudinal Load-Bearing Tissue Radiomic plus Biochemical biomarker and Clinical variable Model (LBTRBC-M) to predict KOA progression.

### Methods and findings

Data from the Foundation of the National Institutes of Health Osteoarthritis Biomarkers Consortium were used. We selected 594 participants with Kellgren-Lawrence grades 1–3 and complete biomarker data. The mean age was 61.6 ± 8.9 years, 58.8% were female, and the racial distribution was 79.3% White or White, 18.0% Black or African American, and 2.7% Asian or other non-White. A total of 1,753 knee MRIs were included across the study period, comprising 594 at baseline, 575 at 1-year follow-up, and 584 at 2-year follow-up. Outcomes included (1) both Joint Space Narrowing (JSN) and pain progression ($n = 567$), (2) only JSN progression ($n = 303$),

**Data availability statement:** The data that support the findings of this study are publicly available through the Osteoarthritis Initiative (OAI) repository at https://nda.nih.gov/oai/. De-identified patient-level clinical data, outcome data, and MRI imaging data used in this study can be accessed from this repository. The specific dataset utilized is clearly identifiable upon accessing the repository. Additionally, the source code for the predictive model developed in this study is available at https://github.com/dmlc/xgboost, and a permanently archived version has been deposited in Zenodo at https://doi.org/10.5281/zenodo.15680828.

**Funding:** This work was supported by the National Key Research & Development Program of China (2023YFE0209700, C.H.D. was funded; https://service.most.gov.cn/), the National Natural Science Foundation of China (Grant No. 82373653, C.H.D. was funded, 82103903, C.H.D. was funded, 82002344, T.W. was funded; https://www.nsfc.gov.cn/), Science and Technology Projects in Guangzhou (Grant No. SL2023A04J02586, C.H.D. was funded; https://kjj.gz.gov.cn/), and the Key development projects of the Sichuan Provincial Science and Technology Plan (Grant No. 2024YFFK0298, S.F.L. was funded; https://kjt.sc.gov.cn/) and the Chengdu Medical Research Project (Grant number: 2024487, S.F.L. was funded; https://cdwjw.chengdu.gov.cn/). The funders had no role in study design, data collection and analysis, decision to publish, or preparation of the manuscript.

**Competing interests:** I have read the journal's policy and the authors of this manuscript have the following competing interests: D.H. is the editor of the osteoarthritis section for UpToDate and co-Editor in Chief of Osteoarthritis and Cartilage. D.H. provides consulting advice on scientific advisory boards for Haleon, TLCBio, Novartis, Tissuegene, Sanofi, Enlivex.

**Abbreviations :** GBD, global burden of disease; JSN, joint space narrowing; KOA, knee osteoarthritis; OAI, osteoarthritis initiative; TBT, trabecular bone texture; TKR, total knee replacement.

(3) only pain progression ($n = 295$), and (4) non-progression (JSN or pain) ($n = 588$), corresponding to an approximate ratio of 2:1:1:2. JSN progression was defined as a minimum joint space width (JSW) loss of ≥0.7 mm, and pain progression as a sustained (≥2 time points) increase of ≥9 points on the Western Ontario and McMaster Universities Osteoarthritis Index (WOMAC) pain subscale (0–100 scale). Using the eXtreme Gradient BOOSTing (XGBOOST) algorithm, the model was developed in the total development cohort ($n = 877$) and tested in the total test cohort ($n = 876$). In the total test cohort, the Area Under the receiver operating characteristic Curve (AUC) of LBTRBC-M for predicting JSN and pain progression, JSN progression, pain progression, and non-progression were 0.880 (95% confidence interval (CI) [0.853, 0.903]), 0.913 (95% CI [0.881, 0.937]), 0.886 (95% CI [0.856, 0.910]), and 0.909 (95% CI [0.888, 0.926]), respectively. The overall accuracy of LBTRBC-M was 70.1%. With LBTRBC-M assistance, the prognostic accuracy of resident physicians ($n = 7$) improved from 44.7%–49.0% to 64.4%–66.5%. The main limitations include the use of a non-routine MRI sequence, the lack of external validation in independent cohorts, and limited incorporation of all knee joint structures in radiomic feature extraction.

## Conclusions

In this study, we observed that longitudinal MRI radiomic features of load-bearing knee joint tissues provide potentially informative markers for predicting knee osteoarthritis progression. These findings may help guide future efforts toward early risk stratification and personalized management of KOA.

### Author summary
#### Why was this study done?

- Knee osteoarthritis (KOA) is a progressive disease with both structural and symptomatic worsening.

- There is currently no established predictive model that integrates longitudinal MRI radiomic features, biochemical biomarkers, and clinical variables to forecast KOA progression.

- Early prediction of KOA progression could support timely intervention and personalized management.

#### What did the researchers do and find?

- We analyzed data from 594 participants with Kellgren-Lawrence grades 1-3 in the FNIH Osteoarthritis Biomarkers Consortium dataset, incorporating 1,753 knee MRIs over a 2-year period.

- We developed a predictive model, the Load-Bearing Tissue Radiomic plus Biochemical biomarker and Clinical variable Model (LBTRBC-M), using the XGBOOST algorithm.

- The model achieved high accuracy in predicting KOA progression outcomes in an independent test cohort, with AUCs ranging from 0.880 to 0.913.

- The use of LBTRBC-M improved the prognostic accuracy of seven resident physicians from 44.7%–49.0% to 64.4%–66.5%.

## What do these findings mean?

- Longitudinal MRI radiomic features of load-bearing knee joint tissues, when combined with biomarkers and clinical data, may help identify patients at higher risk of KOA progression.

- These results support the potential clinical utility of AI-assisted prediction tools for enhancing early diagnosis and individualized treatment planning in KOA.

- The model still needs to be tested in other groups of patients and should include more parts of the knee joint to make sure it works well for different people and in real-world clinical settings.

## Introduction

Osteoarthritis (OA) is a prevalent articular disease that can cause chronic joint pain and debilitating symptoms, significantly impacting patients' quality of life [1,2]. With the aging population and increasing risk factors, the global incidence of OA is expected to rise [3]. The Global Burden of Disease (GBD) project estimates a staggering 303.1 million cases of OA worldwide [4]. Unfortunately, no approved Disease-Modifying Osteoarthritis Drugs (DMOADs) are available, making OA a severe condition with unmet medical needs [5]. It is worth noting that approximately 20%–30% of Knee Osteoarthritis (KOA) patients may progress to end-stage disease, necessitating Total Knee Replacement (TKR) [6].

The knee is the most commonly affected weight-bearing joint in older adults. In KOA patients, excessive stress and strain on load-bearing tissues, encompassing bone, cartilage, and meniscus, can also lead to cartilage deterioration and microscopic bone damage [7]. Some models were developed using knee Magnetic Resonance Image (MRI) and biochemical markers to predict KOA progression in the Foundation for the National Institutes of Health (FNIH) OA Biomarkers Consortium study [8–12]. Combining quantitative and semiquantitative data of load-bearing tissues with biochemical biomarkers, models achieved an Area Under the receiver operating characteristic Curve (AUC) of 0.641–0.722 for predicting Joint Space Narrowing (JSN) and pain progression [8]. Using semiquantitative change of cartilage and meniscus data, models achieved AUCs of 0.706–0.740 [9]. Predictive models based on radiographic subchondral Trabecular Bone Texture (TBT) had AUC of 0.633–0.649 [10]. Finally, models incorporating urinary C-terminal cross-linked telopeptides of type II collagen (CTX-II), serum hyaluronan, and serum N-telopeptide of type I collagen (NTX-I) had an AUC of 0.667 for predicting JSN and pain progression [12].

Recent advancements by Saarakkala and colleagues [13] and Lespessailles and colleagues [14] have demonstrated the potential of integrating imaging biomarkers with biochemical and clinical data. Saarakkala and colleagues highlighted the utility of deep learning on structural MRI for KOA progression prediction, emphasizing the power of data-driven representations over manually extracted biomarkers [13]. Lespessailles and colleagues underscored the role of trabecular bone texture and multimodal biomarker integration for improved predictive accuracy [14]. These studies highlight the growing recognition of combining multiple biomarker types to enhance precision in KOA predictions.

To date, the MRI semiquantitative data, biomarkers, and KOA symptom scores are widely used to evaluate the clinical benefit of KOA patients. However, a predictive model integrating data from longitudinal MRI radiomics, serum or urine

biomarkers, and clinical high-risk factors is unavailable. In this study, using the Convolutional Neural Network (CNN) algorithm [15], we automatically segmented the knee MRI of load-bearing tissue, including femur, tibia, femorotibial cartilage and meniscus [16,17]. Then, using the eXtreme Gradient BOOSTing (XGBOOST) algorithm, we developed and tested the Load-Bearing Tissue Radiomic plus Biochemical biomarker and Clinical variable Model (LBTRBC-M), incorporating the 2-year follow-up of load-bearing tissue MRI radiomics, biochemical biomarkers, and clinical variables, to predict KOA progression within the subsequent 2 years in the FNIH OA Biomarkers Consortium cohort study.

## Patients and methods

### Study design and participants

The current study used publicly available, de-identified data from the Osteoarthritis Initiative (OAI), an ongoing, multi-center, prospective cohort study (Clinical Trials.gov identifier: NCT00080171). As such, no additional ethical approval was required for secondary analysis. The original OAI study protocol was approved by the institutional review boards of all participating centers, including the coordinating center at the University of California, San Francisco (IRB number: 10-00532). The study was conducted in accordance with the Health Insurance Portability and Accountability Act (HIPAA), and all participants provided informed consent.

Six-hundred participants were selected based on frequent knee pain and a Kellgren-Lawrence Grade (KLG) of 1, 2, or 3 on knee radiographs at baseline [18]. They were required to have the baseline and 24-month radiographic data on medial tibiofemoral joint space width (JSW) [19], knee MRI, stored serum and urine specimens, and clinical data. Further details can be found in our study's flowchart (S1 Fig).

### Inclusion and exclusion criteria

FNIH OA Biomarkers Consortium undertook a nested case-control study (194 JSN and pain progression cases and 406 OA comparators) of progressive KOA within the OAI, a unique longitudinal cohort with a publicly available repository of joint images, biologic specimens, and clinical data obtained at annual clinic visits. Details of the study design have been published previously [18]. Briefly, participants eligible for the present study were those who had at least 1 knee with a KLG of 1–3 at baseline determined at a central reading site and for whom knee radiographs, knee MRIs, stored serum and urine specimens, and clinical data were available for the baseline and 24-month visits. One index knee was selected for each participant.

Knees were excluded from analysis under the following circumstances: if progression criteria were met by 12 months to enable the study of change in biomarkers before the progression definition was met, if radiographic lateral joint space narrowing (JSN) grade 2 or 3 was present at baseline, or if total knee replacement or total hip replacement had occurred prior to 24 months due to possible effects on biochemical markers.

### Clinical outcome

A predetermined number of index knees were categorized into four groups based on outcome assessment at 24 months: (1) knees with both JSN and pain progression ($n = 194$), (2) knees with JSN progression but not pain progression ($n = 103$), (3) knees with pain but not JSN progression ($n = 103$), and (4) knees with neither JSN nor pain progression ($n = 200$). The main analysis focused on comparing knees with both JSN and pain progression to all other knees [12,20]. JSN progression was defined as a minimum JSW loss of ≥0.7 mm, and pain progression as a sustained (≥2 time points) increase of ≥9 points on the Western Ontario and McMaster Universities Osteoarthritis Index (WOMAC) pain subscale (0–100 scale) [18].

After enrollment, participants underwent routine clinical examination, knee MRI, and knee joint radiography every 12 months for 2 years.

## Clinical examination and radiography

The FNIH OA Biomarkers Consortium cohort study used the following clinical examination: [1] Knee pain severity scale. [2] Participant global assessment. [3] WOMAC Osteoarthritis Index. [4] Knee Outcomes in Osteoarthritis Survey (KOOS). [5] Limitation in activity due to knee pain. [6] General health and functional status. [7] Walking ability and endurance. [8] Upper leg strength. [9] Serum and urine biochemical biomarkers assay.

The knee fixed-flexion, posterior-anterior weight-bearing radiographs obtained at baseline, and all annual follow-up visits were performed using a Plexiglas positioning frame (SynaFlexer; BioClinica, Newark, CA), with knees flexed to 5–15° and feet internally rotated 10°. All radiographs were centrally read to determine the KLG [21]. The KLG assessments for radiographs were performed by Dr. Piran Aliabadi, MD and Dr. Burt Sack, MD, under the direction of Dr. David Felson, MD from the Boston University Clinical Epidemiology Research and Training Unit for the baseline through 24-month visits. Specifically, Dr. Piran Aliabadi and Dr. Burt Sack are board-certified radiologists with extensive experience in musculoskeletal imaging, and the assessments were conducted under the supervision of Dr. David Felson, a senior rheumatologist and epidemiologist with decades of expertise in osteoarthritis research at the Boston University Clinical Epidemiology Research and Training Unit. The minimum JSW in the medial femorotibial compartment was measured using automated software [22].

## Biochemical biomarkers

Eighteen serum biomarkers included Cartilage Oligomeric Matrix Protein Cartilage Oligomeric Matrix Protein (sCOMP), Hyaluronic Acid (sHA), Type IIA Procollagen Amino-terminal Propeptide (sPIIANP), Type I Collagen C-terminal Telopeptide (sCTX-I), Aggrecan Chondroitin Sulfate 846 Epitope (sCS846), Matrix Metalloproteinase-3 (sMMP-3), Cleavage neoepitope of type II collagen (sC2C), Type II Collagen Neoepitope (sC1, 2C), C-Propeptide of type II collagen (sCPII), sNTX-I, and Nitrated triple helix of type II collagen (sColl2−1NO2). Urine biomarkers included C-terminal cross-linked telopeptide of type I collagen (α-isomer) (uCTX-Iα), C-terminal cross-linked telopeptide of type I collagen (β-isomer) (uCTX-Iβ), uNTX-I, Cleavage neoepitope of type II collagen (uC2C), Type II collagen neoepitope (uC1, 2C), Nitrated triple helix of type II collagen (uColl2−1NO2), and uCTX-II. These markers reflect processes such as cartilage degradation and synthesis, bone turnover, and joint inflammation [12]. Serum and/or urine biochemical markers were measured, and urinary markers were standardized based on urinary creatinine concentration. The interassay coefficients of variation for these markers ranged from 3% to 12% [12].

## MRI protocol and assessment

This study is based on baseline SAGittal 3-Dimensional Double Echo Steady-State with selective Water Excitation (SAG-3D-DESS-WE) MRI data acquired by the FNIH OA Biomarkers Consortium cohort study using Siemens Trio 3.0 Tesla scanners (Magnetom Trio, Siemens Healthcare, Erlangen, Germany) [23]. The SAG-3D-DESS-WE series utilizes near anisotropic voxels (0.7 mm slice thickness × 0.37 mm × 0.46 mm) to maximize in-plane sagittal spatial resolution in a reasonable acquisition time (10.5 min). The protocol of SAG-3D-DESS-WE can be found in S1 Table.

Two experienced musculoskeletal radiologists (F. Roemer and A. Guermazi) independently assessed the MRIs according to Magnetic resonance imaging OsteoArthritis Knee Score (MOAKS). All scores showed substantial (0.61–0.8) or high (0.81–1.0) agreements regarding intra-observer and inter-observer reliabilities.

An automated MRI segmentation approach using CNN [15] was developed for six anatomical structures (femur, tibia, femoral and tibial cartilages [16], and both meniscus [17]). To compute features that might be suitable for being used as biomarkers for KOA progression, we employ methods of automatic image segmentation to specify the anatomical Volume of Interests (VOIs) using CNN. Ambellan and colleagues [16] used the method to segment the femur, tibia, femoral, and tibial cartilage. The method of Tack and colleagues [17] was used to segment the medial and lateral meniscus.

VOIs ($n = 20$ knees) defining the femur, tibia, femorotibial cartilages, and meniscus were manually adjusted by two independent authors (including S.F.L., with 6 years of experience in orthopedics), all of whom were blinded to the clinical outcome data. The segmentation and MRI were viewed and adjusted using itk-SNAP version 3.8.0 software (www.itksnap.org). The Dice Similarity Coefficient (DSC) was used to assess manual adjustment and automated segmentation agreement (S4 Table).

### Double echo steady-state signal feature map of load-bearing tissues

We developed the feature maps of load-bearing tissues using their own Double Echo Steady-State (DESS) signal intensity's (mean pixel value), which builds upon previous studies that visualized the texture of knee MRI [24–26]. We used this method to detect the MRI changes in KOA progression participants.

### Model development and test

In the FNIH OA Biomarkers Consortium cohort study, excluding 47 knee MRIs with non-conforming MR images, there were 594 participants with 1,753 knee MRIs selected during a 2-year follow-up. The knee MRIs were randomly split into a development cohort and a test cohort with a ratio of 1:1 at each visit time point. Each time point (−100 samples per group) would yield only −20 samples with an 8:2 or 7:3 split. A 1:1 split ensured better representation, enhancing model reliability. Although the FNIH dataset was originally constructed as a nested case-control study with matched pairs, we disrupted the original matching during the cohort split. Specifically, participants were randomly assigned to the development and test cohorts, disregarding their original case-control pairings. This strategy allowed us to evaluate model performance in a setting more reflective of real-world clinical variability and supported the use of generalized learning approaches, rather than those relying on strict pairwise comparison. The ratio of JSN and pain progression, JSN progression, pain progression, and non progression groups was 2:1:1:2 in each cohort.

The predictive model was developed in a total development cohort ($n = 877$), which was integrated by the development cohort of three visits. This model was tested in a total test cohort ($n = 876$), which incorporated by test cohort of three visits, including test cohort 1 ($n = 301$) at baseline, test cohort 2 ($n = 297$) at 1-year follow-up, and test cohort 3 ($n = 278$) at 2-year follow-up (Fig 1). The predictive performance was validated using 10-fold cross-validation (Repeated 100 interactions). To assess model stability, we compared LBTRBC-M performance across 100, 300, 1,000, and 25,000 cross-validation

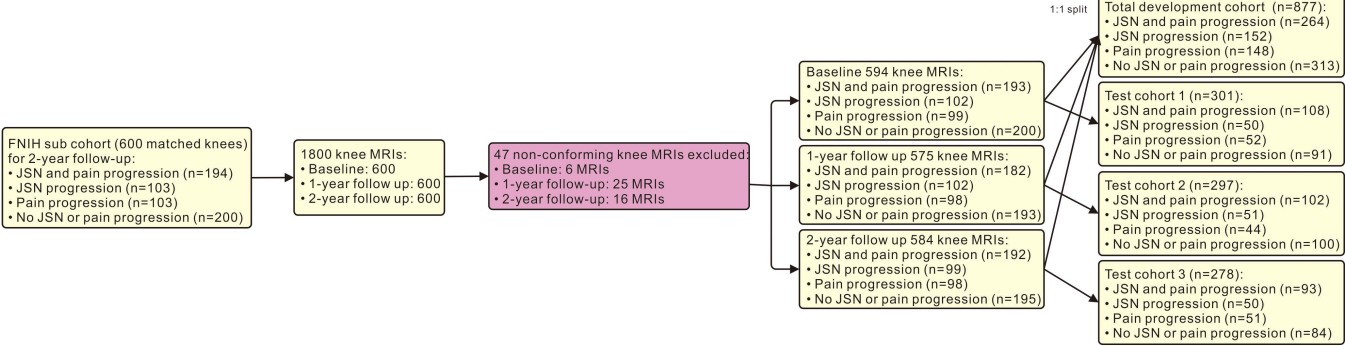

**Fig 1. Participant flow and MRI timeline.** A total of 600 eligible knees were identified from the FNIH OA Biomarkers Consortium cohort. After excluding 47 non-conforming knee MRIs, 1,753 knee MRIs were included over the 2-year follow-up period: 594 at baseline, 575 at 1-year follow-up, and 584 at 2-year follow-up. The predictive model was developed by total development cohort (877 knee MRIs) and tested by total test cohort (876 knee MRIs) which both were incorporated by three visits: baseline, 1-year follow-up, and 2-year follow-up. The ratio between the development cohort and test cohort was 1/1 in each visit time point. FNIH OA Biomarkers Consortium cohort: Foundation of the National Institutes of Health Osteoarthritis Biomarkers Consortium cohort, MRI: Magnetic Resonance Image, JSN: Joint Space Narrowing.

iterations (S18 Table). Given the limited performance gain and the high computational cost, we chose 100 iterations for the final analysis as a practical balance between accuracy and efficiency. The specifications of our hardware specs were Intel Core i5-1135G7 Central Processing Unit 2.40 GHz, 16 GB Random access memory, and a 512 GB Solid-State Drive.

### Three-dimensional MRI radiomic feature analysis

LBTRBC-M was developed in the total development cohort by integrating MRI radiomic features from load-bearing tissues, biochemical biomarkers, and clinical high-risk variables (Fig 2). The MRI protocol (S1 Table) and automatic CNN-based segmentation scheme (S2 Fig) were shown in our study. Image Biomarker Standardisation Initiative (IBSI) standardized MRI radiomic features [27] were extracted using Standardized Environment for Radiomics Analysis (SERA) package [28] from baseline, 1-year follow-up, and 2-year follow-up sagittal SAG-3D-DESS-WE MRI, and predictive models were constructed using an XGBOOST algorithm after using Least Absolute Shrinkage and Selection Operator (LASSO) logistic regression (Repeated 1,000,000 times) for feature selection (S4 and S5 Figs). We have included a comparison of the number of features before and after LASSO selection. Initially, 2,947 features were considered in the model: 2,922 MRI radiomic features, 17 biochemical biomarkers, and 7 clinical variables. After applying LASSO, 255 non-zero features were retained in the final model, including 236 MRI radiomic features, 13 biochemical biomarkers, and 6 clinical variables. In our analysis, we utilized Statistical Analysis System software (SAS, version 9.4), leveraging the generalized regression platform with the LASSO method. The selection of the optimal $\lambda$ was guided by the adaptive weighting approach combined with the Akaike Information Criterion, corrected (AICc) validation method. Specifically, $\lambda$ was chosen from the optimal solution path where the AICc value was minimized, ensuring the best balance between model fit and complexity. The assumptions underlying LASSO regression, including the distribution of errors and the linearity of relationships between predictors and the outcome, were assessed during model development. The linearity assumption was checked through exploratory data analysis and transformations applied to variables as necessary. Residual diagnostics were performed to confirm that the error terms were approximately normally distributed. In addition to LBTRBC-M, several other models, including the Load-Bearing Tissue Radiomic Model (LBT-RM), Femur Radiomic Model (FE-RM), Femoral Cartilage Radiomic Model (FC-RM), Tibia Radiomic Model (TI-RM), Tibial Cartilage Radiomic Model (TC-RM), Lateral Meniscal Radiomic Model (LM-RM), Medial Meniscal Radiomic Model (MM-RM), MOAKS models, Biochemical biomarker Model (BM), and Clinical Model (CM) were constructed for comparison.

### Selection of hyperparameters, model fitting and optimization process

The hyperparameters for XGBOOST and LASSO were selected based on prior literature and initial experiments. Specifically, the values for XGBOOST (max_depth: 6, subsample: 1, colsample_bytree: 1, min_child_weight: 1, $\alpha$: 0, $\lambda$: 1, learning_rate: 0.3, iterations: 100) were chosen in accordance with recommendations in previous studies and through empirical testing to optimize model performance (S16 Table). These settings were further refined using grid search (150 grid points) with cross-validation to identify the best-performing combination. For LASSO, the penalty parameter ($\lambda$) was optimized using cross-validation to minimize the mean squared error (MSE). The selection of hyperparameters was guided by Hastie and colleagues [29] and adapted to the specifics of the dataset and model requirements.

In this study, we used XGBOOST with 10-fold cross-validation to fit and optimize the model. Model performance was assessed using metrics such as AUC, accuracy, Logloss, and Root Average Squared Error (RASE). Key hyperparameters (S15 Table), including max_depth, learning_rate, subsample, and $\lambda$, were optimized via grid search with cross-validation, with practical constraints based on prior literature (e.g., max_depth between 3 and 10, learning_rate between 0.1 and 0.3).

### Reader experiments

Seven resident physicians (Clinical practice years: 1–4 years, physicians list: Liu, Zhao, Cao, J Li, Chen, X Wang, Dang, M Zhang) participated in a study to predict the progression of KOA using knee MRI (SAG-3D-DESS-WE sequence), biochemical biomarkers (17 types of serum and urine biomarkers), and clinical variables (Age, sex, Body Mass Index (BMI),

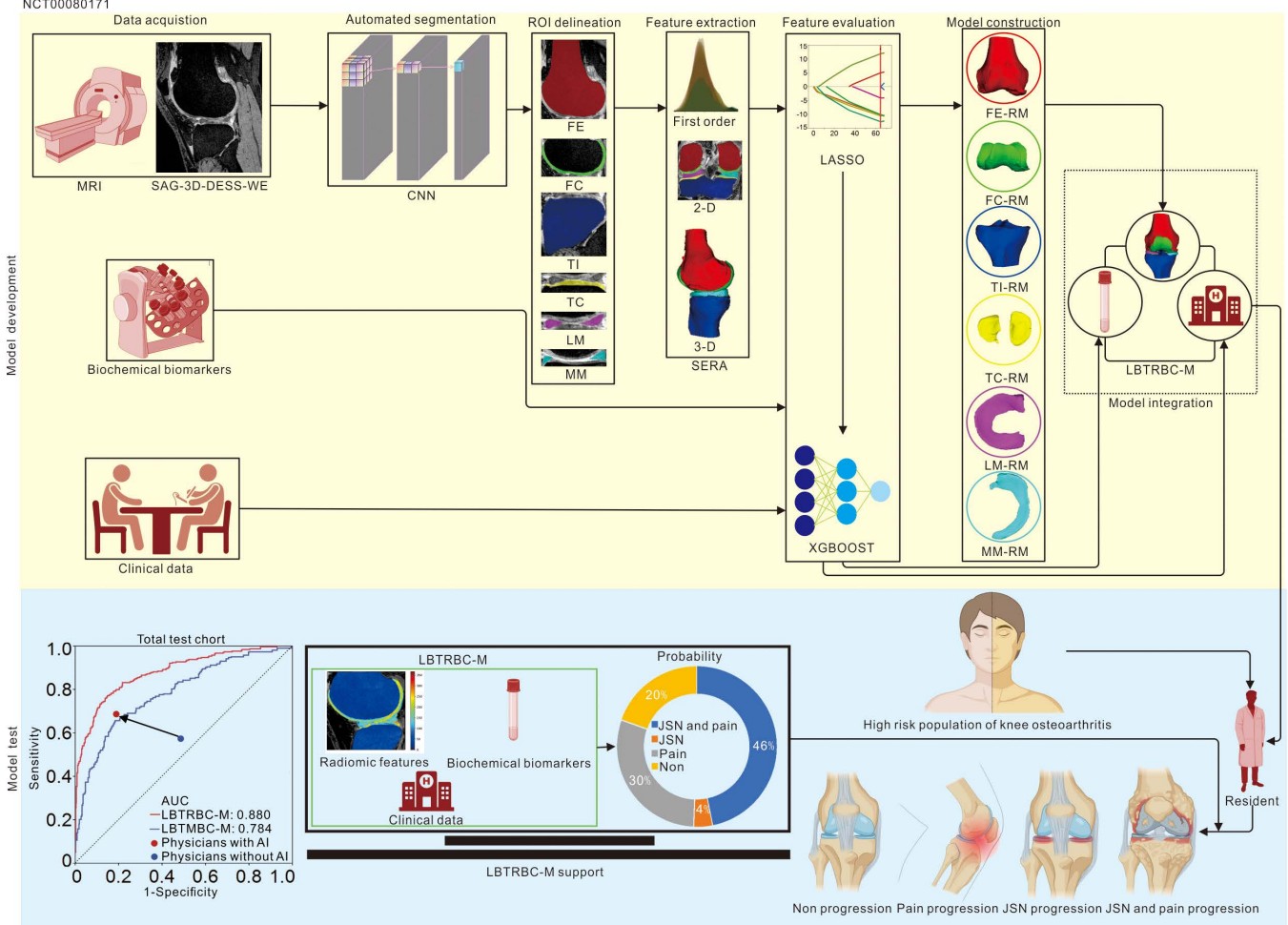

**Fig 2. Workflow of the study.** Knee MRIs of the FNIH OA Biomarkers Consortium cohort were automatically segmented by CNN for feature extraction. After feature evaluation and modeling using the LASSO and XGBOOST algorithm, respectively, eight types of features (biochemical biomarkers, clinical variables, FE, FC, TI, TC, LM, and MM MRI radiomic features) were generated and further used to develop the LBTRBC-M. The performance of LBTRBC-M in predicting KOA progression (i.e., JSN and pain progression vs. JSN progression vs. pain progression vs. non progression) was tested in three visits. FNIH OA Biomarkers Consortium cohort: Foundation of the National Institutes of Health Osteoarthritis Biomarkers Consortium cohort, SAG-3D-DESS-WE: Sagittal 3-Dimensional Double Echo Steady-State with selective Water Excitation, CNN: Convolutional Neural Network, XGBOOST: eXtreme Gradient BOOSTing, LASSO: Least Absolute Shrinkage and Selection Operator, FE-RM: Femur Radiomic Model, FC-RM: Femoral Cartilage Radiomic Model, TI-RM: Tibia Radiomic Model, TC-RM: Tibial Cartilage Radiomic Model, LM-RM: Lateral Meniscal Radiomic Model, MM-RM: Medial Meniscal Radiomic Model, LBTRBC-M: Load-Bearing Tissue Radiomic plus Biochemical biomarker and Clinical variable Model, JSN: Joint Space Narrowing, SERA: Standardized Environment for Radiomics Analysis, MRI: Magnetic Resonance Image.

race, WOMAC pain score, WOMAC disability score, and pain medication use). The study also evaluated the use of LBTRBC-M in assisting the predictions. The LBTRBC-M system provided the probability of four clinical outcomes (JSN and pain progression, JSN progression, pain progression, and non-progression). We used color coding to differentiate prediction results within various probability ranges. Since the outcome variable was a four-class label, random prediction would assign approximately 25% probability to each class. Therefore, when the LBTRBC-M model output a probability above 25.0% for a given class, we considered this as indicative of stronger prediction confidence and displayed the results were displayed in red font; otherwise, they were shown in black font. This color-coding scheme was designed to help resident physicians better understand and utilize the model's output.

## Evaluation of model performance

We evaluated the accuracy of the XGBOOST MRI radiomic models in the prediction of KOA progression using ROC analysis. Evaluation metrics were computed, including the Area Under ROC Curve (AUC), sensitivity, specificity, and kappa value. In this study, a "successful prediction" is defined through a combination of predictive accuracy and clinical relevance. From a statistical perspective, a prediction is considered successful if the model achieves an AUC greater than 0.70, a widely accepted threshold indicating reliable discriminative performance for clinical decision-making [30]. Beyond statistical accuracy, clinical utility serves as a crucial measure of success. A prediction is clinically meaningful if it helps identify high-risk patients likely to experience JSN progression and/or pain progression within the next two years. Furthermore, the model's ability to integrate multiple biomarkers and clinical variables enhances its capacity to provide a comprehensive risk assessment, ensuring its applicability in real-world clinical decision-making. The performance of the predictive model was validated using 10-fold cross-validation (S3 Fig).

## Missing data

Missing data were addressed using multiple imputation (Predictive mean matching). Specifically, data were missing for 2 knees in the MOAKS femur lateral posterior bone marrow lesion percentage (lesion that is edema), 13 knees for serum biomarkers, and 10 knees for urine biomarkers. The percentages of missing data were 0.3%, 2.1%, and 1.7%, respectively. Given these low percentages, the impact on the results was deemed minimal.

## Statistical analysis

Generalized Estimating Equation (GEE) was used to assess the risk of KOA progression adjusting for the confounders (age, sex, BMI, knee side, race, WOMAC knee pain score) [31–34]. These confounders were selected based on prior literature and their potential impact on KOA progression. The GEE model assumed an autoregressive correlation structure, as this was deemed most suitable for the nature of the repeated measures in our data. MRI radiomic feature extraction and DSC calculation were performed in Matlab R2021a (version 9.10.0). GEE, LASSO logistic regression, XGBOOST modeling, and the Delong test were performed in the SAS. Model performance was assessed using AUC, sensitivity, specificity, accuracy, and kappa value in all development and test cohort. We compared all cohorts using the unpaired $t$ test/one-way ANOVA tests for continuous variables and the $\chi^2$ test, Mann–Whitney test, or Kruskal–Wallis test for categorical variables, as appropriate. For continuous variables, parametric tests (e.g., unpaired $t$ test, one-way ANOVA) were employed when data were normally distributed, as assessed using the Shapiro–Wilk test. Non-parametric tests (e.g., Mann–Whitney test, Kruskal–Wallis test) were used when the assumption of normality was not met. For categorical variables, $\chi^2$ tests were used to compare proportions across groups. To control for type I error in our analysis, both Tukey's honestly significant difference (HSD) test and Bonferroni correction were applied during one-way ANOVA tests for multiple comparisons. Area differences between two Receiver Operator Characteristic (ROC) curves were compared using the DeLong test. For Bonferroni correction, the adjusted $P$ value threshold was calculated by dividing 0.05 by the number of comparisons. Accordingly, we considered results statistically significant if the adjusted $p$ value was below this threshold. The exact adjusted significance levels are reported where relevant. A $P$ value < 0.05 was considered statistically significant unless otherwise specified after correction. R (version 4.1.1) with the pROC package (version 1.18.0) was used for all analyses.

# Results

## Baseline characteristics

At the baseline visit, 293 (178/293, 61% females) individuals were included in the development cohort 1, while 301 (171/301, 57% females) individuals were included in the test cohort 1 (S2 Table). Results of baseline characteristics showed no significant differences between the two cohorts (S2 Table).

Baseline characteristics among the four groups (JSN and pain progression, JSN progression, pain progression, and non-progression) were generally similar, except for age (*p*=0.018), female (*p*=0.016), and KLG (*p*=0.003) in test cohort 1 (Table 1). In addition, the levels of baseline serum/urine biochemical biomarkers showed no significant differences between the four groups in both cohorts (S3 Table).

## Reliability of automatic segmentation

The CNN segmentation dataset was utilized for extracting MRI radiomic features, and all DSCs were above 0.800 (S4 Table). The feature selection process using LASSO regression was displayed in S4 and S5 Figs, and the selected features can be found in S5 Table.

## Feature maps of load-bearing tissues

Fig 3 illustrates the distinct DESS signal intensity changes observed among the four groups across load-bearing structures. The high values of the femur and tibia were detected in JSN and pain progression, and pain progression group. The high values of femoral cartilage, tibial cartilage, lateral meniscus, and medial meniscus were detected in JSN and pain progression, and JSN progression group. As shown in S6 Fig, in the femur, tibia, and medial meniscus, the JSN and pain progression group exhibited the highest MRI radiomic values, followed by the pain progression group, and then the JSN progression group. Conversely, in the femoral cartilage, tibial cartilage, and lateral meniscus, the JSN and pain progression group had the highest MRI radiomic values, followed by the JSN progression group, and then the pain progression group.

**Table 1. Baseline characteristics of participants in development cohort 1 and test cohort 1.**

| | Development cohort 1 (*n*=293) | | | | | Test cohort 1 (*n*=301) | | | | |
| | JSN and pain (*n*=85) | JSN (*n*=52) | Pain (*n*=47) | Non (*n*=109) | *p* value | JSN and pain (*n*=108) | JSN (*n*=50) | Pain (*n*=52) | Non (*n*=91) | *p* value |
|---|---|---|---|---|---|---|---|---|---|---|
| Age[a] | 62.3±9.2 | 63.2±8.2 | 61.1±8.2 | 62.0±9.2 | 0.703 | 61.8±8.5 | 63.2±8.6 | 57.9±9.1 | 60.8±9.0 | 0.018 |
| Female[b] | 53 (62%) | 25 (48%) | 29 (62%) | 71 (65%) | 0.361 | 56 (52%) | 21 (42%) | 35 (67%) | 59 (65%) | 0.057 |
| BMI[a] | 30.7±4.5 | 30.0±4.8 | 30.8±4.4 | 30.4±4.7 | 0.826 | 30.7±5.0 | 31.3±4.3 | 31.2±5.6 | 30.6±4.9 | 0.845 |
| PM[b] | 30 (35%) | 11 (21%) | 17 (36%) | 32 (29%) | 0.488 | 33 (31%) | 11 (22%) | 18 (35%) | 23 (25%) | 0.653 |
| Injury[b] | 24 (28%) | 18 (35%) | 19 (40%) | 34 (31%) | 0.694 | 44 (41%) | 22 (44%) | 19 (37%) | 32 (35%) | 0.798 |
| Surgery[b] | 16 (19%) | 9 (17%) | 5 (11%) | 23 (21%) | 0.777 | 19 (18%) | 11 (22%) | 10 (19%) | 13 (14%) | 0.890 |
| KLG[b] | | | | | 0.960 | | | | | 0.003 |
| 1 | 8 (9%) | 8 (15%) | 6 (13%) | 11 (10%) | | 16 (15%) | 6 (12%) | 7 (13%) | 13 (14%) | |
| 2 | 50 (59%) | 25 (48%) | 23 (49%) | 58 (53%) | | 34 (31%) | 21 (42%) | 35 (67%) | 56 (62%) | |
| 3 | 27 (32%) | 19 (37%) | 18 (38%) | 40 (37%) | | 58 (54%) | 23 (46%) | 10 (20%) | 22 (24%) | |
| MJSW[a] | 4.1±1.3 | 3.9±1.2 | 4.0±1.1 | 3.8±1.0 | 0.358 | 3.6±1.4 | 3.6±1.2 | 3.9±1.0 | 4.0±1.0 | 0.109 |
| WOMAC_PS[b] | 1 (0, 3) | 2 (0, 5) | 1 (0, 3) | 1 (0, 4) | 0.219 | 1 (0, 3) | 2 (0, 5) | 1 (0, 2) | 1 (0, 5) | 0.091 |
| WOMAC_SS[b] | 2 (0, 2) | 2 (0, 3) | 1 (0, 2) | 1 (0, 2) | 0.552 | 1 (0, 2) | 1 (0, 3) | 2 (0, 2) | 1 (0, 3) | 0.905 |
| WOMAC_DS[b] | 4 (0, 13) | 3 (0, 15) | 3 (1, 10) | 4 (0, 11) | 0.962 | 6 (1, 15) | 5 (0, 18) | 6 (1, 15) | 2 (0, 16) | 0.401 |

Data are mean ± SD or number (%).

[a] One-way ANOVA tests are used for differences between means.

[b] Kruskal–Wallis tests are used for differences between ranks.

The results of development cohort 1 and test cohort 1 corresponded to baseline follow-up. JSN: Joint Space Narrowing, BMI: Body Mass Index, PM: Pain Medication, KLG: Kellgren and Lawrence Grade, MJSW: Minimum Joint Space Width, WOMAC_PS: Western Ontario and McMaster Universities Arthritis Index Pain Score, WOMAC_SS: Western Ontario and McMaster Universities Arthritis Index Stiffness Score, WOMAC_DS: Western Ontario and McMaster Universities Arthritis Index Disability Score, SD: Standard Deviation.

PLOS Medicine

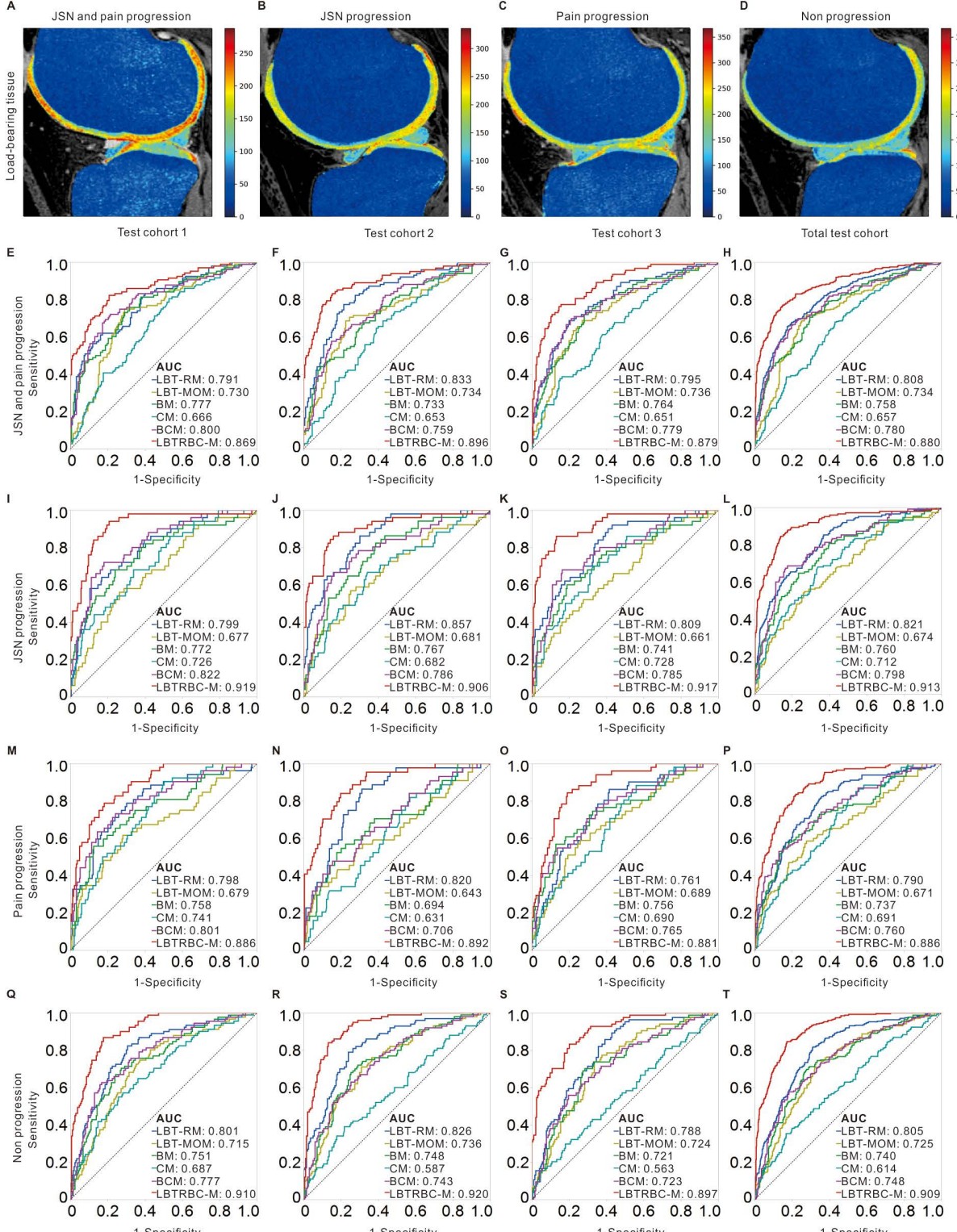

**Fig 3. The DESS signal feature maps of load-bearing tissues in different groups and prediction performance of LBT-RM, LBT-MOM, BM, CM, BCM, and LBTRBC-M in the test cohorts.** DESS signal intensity maps of load-bearing tissues were developed in JSN and pain progression **(A)**, JSN progression **(B)**, pain progression **(C)**, and non progression group **(D)**. The performance of predicting JSN and pain progression **(E–H)**, JSN

progression **(I–L)**, pain progression **(M–P)**, and non progression **(Q–T)** in LBT-RM, LBT-MOM, BM, CM, BCM, and LBTRBC-M in the test cohort 1 to 3 and the total test cohort. The results of test cohort 1, test cohort 2, test cohort 3, and the total test cohort corresponded to baseline, 1-years follow-up, 2-year follow-up, and encompassed the aforementioned follow-up time points. LBT-RM: Load-Bearing Tissue Radiomic Model, LBT-MOM: Load-Bearing Tissue MOAKS Model, BM: Biochemical biomarker Model, CM: Clinical Model, BCM: Biochemical biomarker plus Clinical variable Model, LBTRBC-M: Load-Bearing Tissue Radiomic plus Biochemical biomarker and Clinical variable Model, AUC: Area Under receiver operating characteristic Curve, MOAKS: Magnetic resonance imaging OsteoArthritis Knee Score, DESS: Double Echo Steady-State.

### Algorithm selection

In our study, the performance of the LBTRBC-M model was compared across several machine learning algorithms, including XGBOOST, bootstrap forest, shallow neural network, support vector machines, decision tree, nominal logistic, and naive bayes (S15 Table). Among these, XGBOOST demonstrated the best overall performance, achieving the highest AUC (0.897) and the lowest Root Average Squared Error (RASE) (0.508). Additionally, XGBOOST had a favorable Entropy R Square (ERS) of 0.409 and a low misclassification rate (0.299), outperforming all other algorithms, including bootstrap forest and shallow neural network. Given its superior discriminative ability and error minimization, XGBOOST was selected as the optimal algorithm for predicting KOA progression in this study.

### Performance of single-structure MRI radiomic models: Comparison with MOAKS imaging marker

The Single-Structure Radiomic Models (SS-RMs), including FE-RM, FC-RM, TI-RM, TC-RM, LM-RM, and MM-RM, showed low kappa values ranged from 0.042 to 0.296 (S7 Fig), moderate AUCs ranged from 0.507 (95% confidence interval (CI) [0.424, 0.589]) to 0.747 (95% CI [0.685, 0.801]) (S8–S13 Figs and S6 Table), and low accuracy ranged from 30.9% to 51.5% (S7 Table) in the test cohorts.

In the total test cohort, comparisons with Single-Structure MOAKS Models (SS-MOMs), SS-RMs showed significant AUC improvement in predicting KOA progression, including JSN and pain progression, JSN progression, and pain progression, except for FC-RM versus FC-MOM in predicting JSN and/or pain progression (S8 Table).

### Performance of LBTRBC-M: Comparison with SS-RMs

The integration of SS-RMs into the LBT-RM resulted in improved predictive accuracy. The LBT-RM achieved kappa values ranged from 0.355 to 0.450 (S14 Fig), AUCs ranged from 0.761 (95% CI [0.685, 0.823]) to 0.857 (95% CI [0.796, 0.903]) (Fig 3E–3T), and accuracy ranged from 54.0% to 61.6% (S7 Table) in the test cohorts. In comparison, the Load-Bearing Tissue MOAKS Model (LBT-MOM), BM, CM, and Biochemical biomarker plus Clinical variable Model (BCM) had lower AUC values in the test cohorts, ranging from 0.643 (95% CI [0.541, 0.733]) to 0.736 (95% CI [0.668, 0.794]), 0.694 (95% CI [0.597, 0.776]) to 0.777 (95% CI [0.717, 0.827]), 0.563 (95% CI [0.487, 0.636]) to 0.741 (95% CI [0.669, 0.802]), and 0.706 (95% CI [0.613, 0.784]) to 0.822 (95% CI [0.751, 0.877]), respectively (S6 Table). Furthermore, the kappa values, AUCs, and accuracy of LBTRBC-M ranged from 0.551 to 0.628, 0.869 (95% CI [0.819, 0.907]) to 0.919 (95% CI [0.883, 0.946]), and 68.0% to 74.1% in the test cohorts (Fig 3E–3T and S6 and S7 Tables).

When predicting KOA progression, LBTRBC-M outperformed LBT-RM, BM, CM, BCM, and Load-Bearing Tissue MOAKS Model (LBTMBC-M) with significant AUC differences (Tables 2 and S8, $p < 0.050$). To predict JSN and pain progression, the AUC differences between LBTRBC-M and LBT-RM, BM, CM, BCM, and LBTMBC-M were 0.072 (95% CI [0.044, 0.100]; $p < 0.001$), 0.122 (95% CI [0.088, 0.157]; $p < 0.001$), 0.224 (95% CI [0.183, 0.264]; $p < 0.001$), 0.100 (95% CI [0.067, 0.134]; $p < 0.001$), and 0.096 (95% CI [0.064, 0.128]; $p < 0.001$) in the total test cohort, respectively (Table 2). Additionally, As shown in S8 Table, when predicting KOA progression, there was no significant AUC difference between LBT-RM and BCM in the total test cohort, such as JSN and pain progression, the AUC difference was 0.028 (95% CI [−0.015, 0.072], $p = 0.203$), for JSN progression, 0.023 (95% CI [−0.029, 0.075], $p = 0.379$), and for pain progression, 0.030 (95% CI [−0.026, 0.086], $p = 0.288$).

**Table 2. Comparing the areas under two correlated ROC curves between predictive models in the test cohorts.**

| Predicting models | Test cohort 1 | | Test cohort 2 | | Test cohort 3 | | Total test cohort | |
|---|---|---|---|---|---|---|---|---|
| | AUC difference | p value | AUC difference | p value | AUC difference | p value | AUC difference | p value |
| **JSN and pain progression** | | | | | | | | |
| LBTRBC-M vs. LBT-RM | 0.078 (0.025, 0.131) | 0.004 | 0.063 (0.020, 0.106) | 0.004 | 0.084 (0.036, 0.132) | <0.001 | 0.072 (0.044, 0.100) | <0.001 |
| LBTRBC-M vs. BM | 0.092 (0.034, 0.151) | 0.002 | 0.163 (0.100, 0.226) | <0.001 | 0.115 (0.058, 0.172) | <0.001 | 0.122 (0.088, 0.157) | <0.001 |
| LBTRBC-M vs. Clinical model | 0.204 (0.132, 0.275) | <0.001 | 0.243 (0.176, 0.309) | <0.001 | 0.227 (0.156, 0.299) | <0.001 | 0.224 (0.183, 0.264) | <0.001 |
| LBTRBC-M vs. BCM | 0.070 (0.012, 0.127) | 0.018 | 0.137 (0.078, 0.197) | <0.001 | 0.100 (0.041, 0.158) | <0.001 | 0.100 (0.067, 0.134) | <0.001 |
| LBTRBC-M vs. LBTMBC-M | 0.067 (0.014, 0.121) | 0.014 | 0.119 (0.061, 0.176) | <0.001 | 0.109 (0.053, 0.165) | <0.001 | 0.096 (0.064, 0.128) | <0.001 |
| **JSN progression** | | | | | | | | |
| LBTRBC-M vs. LBT-RM | 0.120 (0.066, 0.175) | <0.001 | 0.049 (−0.009, 0.107) | 0.098 | 0.108 (0.061, 0.155) | <0.001 | 0.092 (0.061, 0.123) | <0.001 |
| LBTRBC-M vs. BM | 0.147 (0.063, 0.231) | <0.001 | 0.139 (0.062, 0.217) | <0.001 | 0.176 (0.080, 0.272) | <0.001 | 0.153 (0.104, 0.202) | <0.001 |
| LBTRBC-M vs. Clinical model | 0.193 (0.121, 0.265) | <0.001 | 0.225 (0.143, 0.307) | <0.001 | 0.189 (0.116, 0.263) | <0.001 | 0.201 (0.158, 0.245) | <0.001 |
| LBTRBC-M vs. BCM | 0.097 (0.030, 0.163) | 0.004 | 0.120 (0.050, 0.190) | <0.001 | 0.132 (0.052, 0.212) | 0.002 | 0.115 (0.074, 0.157) | <0.001 |
| LBTRBC-M vs. LBTMBC-M | 0.111 (0.044, 0.177) | <0.001 | 0.152 (0.078, 0.225) | <0.001 | 0.149 (0.072, 0.227) | <0.001 | 0.136 (0.094, 0.178) | <0.001 |
| **Pain progression** | | | | | | | | |
| LBTRBC-M vs. LBT-RM | 0.088 (0.030, 0.146) | 0.003 | 0.072 (0.022, 0.123) | 0.005 | 0.120 (0.059, 0.181) | <0.001 | 0.096 (0.063, 0.129) | <0.001 |
| LBTRBC-M vs. BM | 0.128 (0.053, 0.202) | <0.001 | 0.199 (0.122, 0.276) | <0.001 | 0.125 (0.046, 0.204) | 0.002 | 0.149 (0.105, 0.193) | <0.001 |
| LBTRBC-M vs. Clinical model | 0.145 (0.081, 0.210) | <0.001 | 0.261 (0.161, 0.362) | <0.001 | 0.190 (0.108, 0.273) | <0.001 | 0.195 (0.148, 0.243) | <0.001 |
| LBTRBC-M vs. BCM | 0.085 (0.020, 0.149) | 0.010 | 0.187 (0.104, 0.270) | <0.001 | 0.116 (0.043, 0.189) | 0.002 | 0.126 (0.084, 0.169) | <0.001 |
| LBTRBC-M vs. LBTMBC-M | 0.133 (0.058, 0.207) | <0.001 | 0.211 (0.123, 0.299) | <0.001 | 0.143 (0.075, 0.211) | <0.001 | 0.160 (0.116, 0.203) | <0.001 |
| **Non progression** | | | | | | | | |
| LBTRBC-M vs. LBT-RM | 0.109 (0.062, 0.155) | <0.001 | 0.094 (0.055, 0.133) | <0.001 | 0.109 (0.066, 0.152) | <0.001 | 0.104 (0.079, 0.128) | <0.001 |
| LBTRBC-M vs. BM | 0.159 (0.097, 0.221) | <0.001 | 0.172 (0.112, 0.232) | <0.001 | 0.176 (0.111, 0.241) | <0.001 | 0.169 (0.134, 0.205) | <0.001 |
| LBTRBC-M vs. Clinical model | 0.223 (0.155, 0.292) | <0.001 | 0.333 (0.260, 0.405) | <0.001 | 0.334 (0.253, 0.415) | <0.001 | 0.295 (0.252, 0.337) | <0.001 |
| LBTRBC-M vs. BCM | 0.133 (0.073, 0.193) | <0.001 | 0.177 (0.116, 0.238) | <0.001 | 0.174 (0.109, 0.239) | <0.001 | 0.161 (0.125, 0.196) | <0.001 |
| LBTRBC-M vs. LBTMBC-M | 0.097 (0.039, 0.155) | 0.002 | 0.151 (0.096, 0.206) | <0.001 | 0.151 (0.092, 0.211) | <0.001 | 0.132 (0.099, 0.165) | <0.001 |

Data are AUC difference (95% CI).

The results of test cohort 1, test cohort 2, test cohort 3, and the total test cohort corresponded to baseline, one year follow-up, two years follow-up, and encompassed the aforementioned follow-up time points. ROC: Receiver Operating Characteristic, AUC: Area Under the ROC Curve, JSN: Joint Space Narrowing, CI: Confidence Interval, LBT-RM: Load-Bearing Tissue Radiomic Model, LBTRBC-M: Load-Bearing Tissue Radiomic plus Biochemical biomarker and Clinical variable Model, BM: Biochemical biomarker Model, BCM: Biochemical biomarker plus Clinical variable Model, LBTMBC-M: Load-Bearing Tissue MOAKS plus Biochemical biomarker and Clinical variable Model, MOAKS: Magnetic resonance imaging OsteoArthritis Knee Score.

## Association of the model output with KOA progression

S9 Table displays the Odds Ratio (OR) of KOA progression for predictive model output in the entire cohort. The output of LBTRBC-M showed the highest ORs of JSN and pain progression in the predictive models. The adjusted OR of JSN and pain progression for LBTRBC-M output was 30.906 (95% CI [22.470, 42.511]); the adjusted OR of JSN progression for LBTRBC-M output was 6.465 (95% CI [4.740, 8.820]); and the adjusted OR of pain progression for LBTRBC-M output was 3.307 (95% CI [2.457, 4.452]).

## Performance of LBTRBC-M-supported resident physicians: Compared with no LBTRBC-M-supported resident physicians

Seven resident physicians predicted the KOA progression using knee MRI, biochemical biomarkers, and clinical data (Fig 4A). Based on the predictive performance of individual physicians (S10 and S11 Tables), we found the assistance of LBTRBC-M significantly improved the accuracy of resident physicians in predicting KOA progression from 46.9% (95% CI [44.7%, 49.0%]) to 65.4% (95% CI [64.4%, 66.5%]) in the total test cohort (S12 Table). In addition, with LBTRBC-M support, the performance of physicians in predicting JSN and pain progression were also improved, with increased sensitivity and specificity of 68.1% (95% CI [66.2%, 70.0%]; $p < 0.050$) and 80.4% (95% CI [78.9%, 81.8%]; $p < 0.050$) in the total test cohort, respectively {compared to 57.5% (95% CI [51.7%, 63.2%]) and 51.8% (95% CI [48.8%, 54.9%]) without LBTRBC-M support, respectively}. Similarly, the prediction performance, including sensitivity and specificity of JSN progression, pain progression, and non progression in physicians, showed improvement with LBTRBC-M assistance. These improvements were validated in different test cohorts of visits (Fig 4B–4I and S15 and S12 Tables).

## Sensitivity analyses

To evaluate the robustness of our findings, sensitivity analyses were performed by re-running the GEE model using alternative correlation structures, including independent, exchangeable, and unstructured. The results were compared across these assumptions, and the analyses confirmed that our findings remained consistent and robust (S13 Table). To assess the impact of missing data on the study's findings, sensitivity analyses were conducted (S14 Table). These included: comparing the results from the multiple imputation approach with those obtained from a complete case analysis (excluding cases with missing data). The results were consistent across both approaches, confirming that the handling of missing data did not significantly alter the study's conclusions. To evaluate model robustness, we performed a sensitivity analysis by varying key hyperparameters (S17 Table) such as max_depth, learning_rate, subsample, and λ. We used Delong test to assess the effect of each parameter on model performance. This analysis aimed to identify the most influential parameters and quantify uncertainty, improving the model's reliability and understanding its behavior in predicting KOA progression.

## Testing of interactions

We tested the significance of interaction terms by comparing AUC differences between model combinations (LBTRBC-M versus LBTRB-M, LBTRBC-M versus LBTRC-M, LBTRC-M versus LBTRC-M) using the DeLong test. The results, presented in S8 Table, showed no significant AUC differences across test cohorts 1, 2, 3, and the total cohort. This indicates that adding different features (biochemical biomarkers, clinical variables) did not significantly improve model performance.

## Probability distribution patterns in LBTRAB-M model predictions

In our contour plot analysis of the LBTRAB-M model (S3C Fig), probability distributions varied across progression categories. JSN and pain progression clustered between 5.0% and 99.0%, JSN progression between 15.0%−25.0% and 55.0%−75.0%, pain progression between 2.0% and 25.0%, and non-progression between 50.0%−99.0%. These patterns

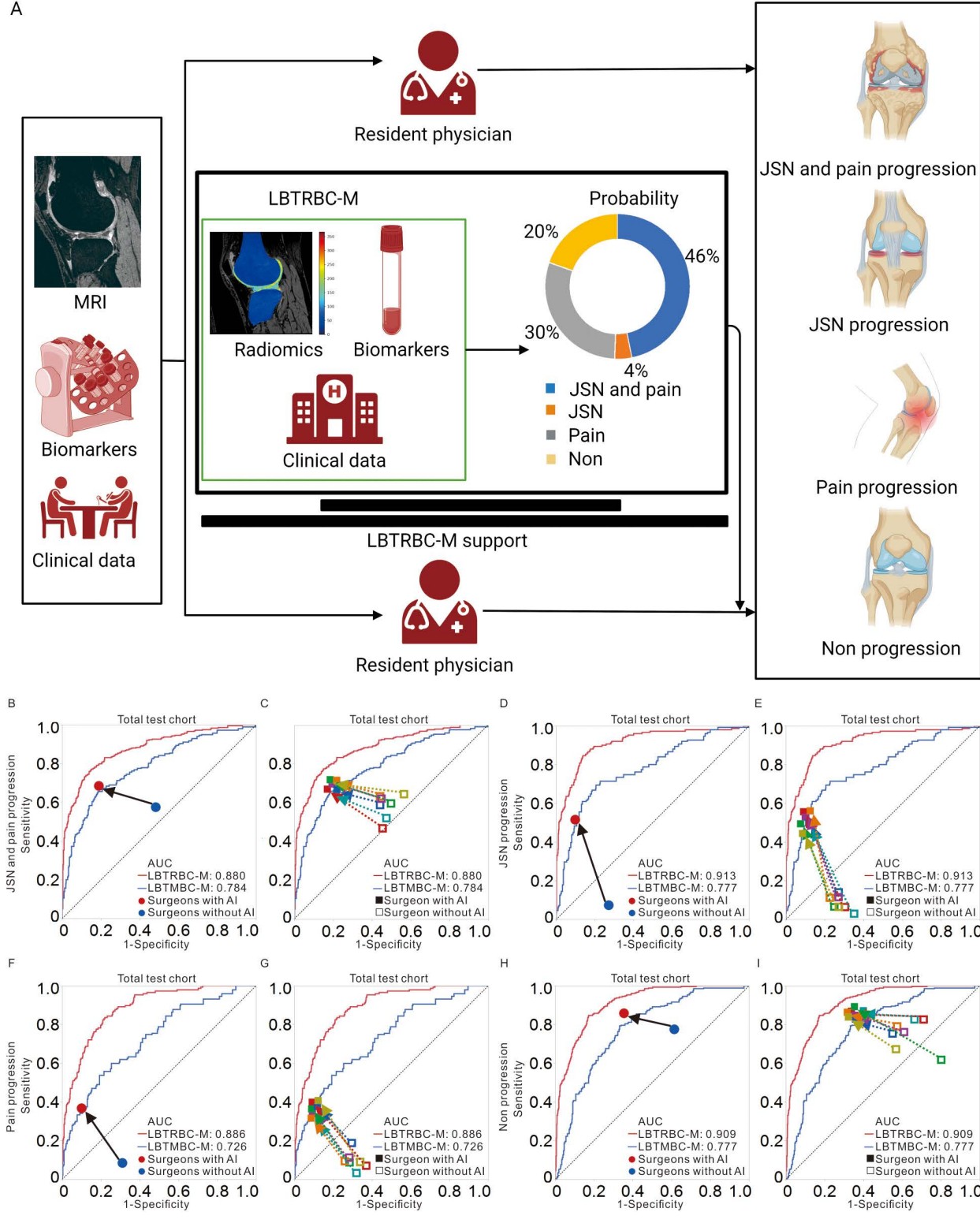

**Fig 4. Performance of resident physicians and models in predicting the KOA progression in the total test cohort.** A schematic workflow of the assessment of KOA progression risk by the resident physicians with LBTRBC-M assistance **(A)**. AUC for predicting JSN and pain progression **(B)**, JSN progression **(D)**, pain progression **(F)**, and non progression **(H)** among the LBTRBC-M, LBTMBC-M, and average performance of all resident physicians

without (blue dot) and with (red dot) the support of LBTRBC-M in the total test cohort was demonstrated. The performance of LBTRBC-M was superior to LBTMBC-M that of clinical experts given baseline MOAKS, serum biochemical level, and clinical data; however, as shown in **B, D, F, and H**, both the sensitivity and specificity of resident physicians were improved when assistance was provided by LBTRBC-M (black arrow). Individual performance of resident physicians was represented by open shapes (without LBTRBC-M support) and filled shapes (with LBTRBC-M support). The prognostic performance of resident physicians was greatly boosted by integrating LBTRBC-M into the loop, as shown in dashed connection lines in **C, E, G, and I**. The results of the total test cohort encompassed the baseline, 1-years follow-up, and 2-year follow-up points. Each colored line represents a different resident physician. AUC: Area Under receiver operating characteristic Curve, LBTRBC-M: Load-Bearing Tissue Radiomic plus Biochemical biomarker and Clinical variable Model, LBTMBC-M: Load-Bearing Tissue MOAKS plus Biochemical biomarker and Clinical variable Model, MOAKS: Magnetic resonance imaging OsteoArthritis Knee Score, AI: Artificial Intelligence.

suggest distinct probability thresholds for each outcome, highlighting the need for ROC and precision-recall curve analyses to optimize sensitivity and specificity for clinical use.

### Stratified cross-validation

To ensure a balanced distribution of outcome types across datasets, we performed a stratified split of the total development and test cohort for the LBTRBC-M model, maintaining an approximate 2:1:1:2 ratio of JSN and pain progression, JSN progression, pain progression, and non-progression (S16A and S16B Fig). Prior to stratification, the AUCs in the total test cohort were: 0.880 (95% CI [0.853, 0.903]) for JSN and pain progression, 0.913 (95% CI [0.881, 0.937]) for JSN progression, 0.886 (95% CI [0.856, 0.910]) for pain progression, and 0.909 (95% CI [0.888, 0.926]) for non-progression. Following stratification, the corresponding AUCs were: 0.853 (95% CI [0.826, 0.877]) for JSN and pain progression ($\Delta$AUC = 0.027, 95% CI [−0.003, 0.058]; $p$ = 0.080); 0.860 (95% CI [0.823, 0.891]) for JSN progression ($\Delta$AUC = 0.053, 95% CI [0.016, 0.090]; $p$ = 0.005), 0.878 (95% CI [0.843, 0.906]) for pain progression ($\Delta$AUC = 0.008, 95% CI [−0.031, 0.047]; $p$ = 0.697), and 0.853 (95% CI [0.824, 0.877]) for non-progression ($\Delta$AUC = 0.056, 95% CI [0.030, 0.082]; $p$ < 0.001) (S16 and S16D Fig and S19 Table).

### Discussion

In this longitudinal and multicenter study, we followed 594 participants, conducting 1,753 knee MRIs over a 2-year follow-up. Our final predictive model, LBTRBC-M, utilized MRI radiomics of load-bearing tissues, biochemical biomarkers, and clinical high-risk factors for KOA to predict the JSN and/or pain progression. The model performed AUCs ranged from 0.869 (95% CI [0.819, 0.907]) to 0.919 (95% CI [0.861, 0.954]), kappa values ranged from 0.551 to 0.628, and accuracy ranged from 68.0% to 74.1% in predicting JSN and/or pain progression in the test cohorts. The output of LBTRBC-M indicated an elevated risk of JSN and pain progression (adjusted OR ranged from 3.307 to 30.906). LBTRBC-M significantly improved the performance of resident physicians in predicting KOA progression, increasing sensitivity, specificity, and accuracy in the test cohorts. The performance of this predictive model was validated in different test cohorts of visits. LBTRBC-M was shown as a potential non-invasive assistance tool for physicians to evaluate whether KOA individuals benefit from personalized decision-making.

Recent studies have advanced KOA progression prediction by integrating MRI radiomics, biochemical biomarkers, and clinical risk factors into predictive models. For example, Saarakkala and colleagues [13] utilized deep learning models on MRI data to predict KOA progression, achieving an AUC of 0.78. Similarly, Lespessailles and colleagues [14] demonstrated the predictive utility of trabecular bone texture from radiographs combined with clinical and biochemical data, emphasizing the value of multimodal integration. Meanwhile, Hu and colleagues [35] developed the DeepKOA model, integrating MRI and clinical data but observed limited improvement when adding clinical features.

Previous research from the FNIH OA Biomarkers Consortium identified MRI and biochemical biomarkers associated with KOA progression [12,36–40]. Previous studies have reported associations between high subchondral bone Trabecular Thickness (Tb. Th) of the medial tibia and KOA progression [36]. Another study found that specific bone phenotypes

were associated with an increased risk of KOA progression [37]. Baseline MRI measurements of bone shape and area in the femur, tibia, and patella were also identified as risk factors for JSN and pain progression [38]. Loss of medial femoro-tibial cartilage thickness was strongly associated with JSN and pain progression, with higher OR values for JSN compared to pain progression [39]. Baseline cartilage volume in the lateral femoral plate predicted medial JSN progression [40]. Additionally, baseline urine levels of CTX-II and CTX-Iα were predictors of JSN and pain progression [12]. In our study, the outputs of LBT-RM, BCM, and LBTRBC-M all increased the odds of KOA progression, including JSN and pain progression, JSN progression, and pain progression, whereas the output of LBTRBC-M performed the highest ORs of KOA progression, with adjusted OR values ranging from 9.938 to 944.796.

Several models have been developed to predict KOA progression in the FNIH OA Biomarkers Consortium study [8,9,12]. However, radiomic models are lacking, particularly in combining longitudinal load-bearing tissue MRI radiomics with biochemical biomarkers. Integrating different image sets and biochemical biomarkers has shown higher predictive accuracy. For instance, using semiquantitative MRI-based cartilage markers alone yielded an AUC of 0.706 for predicting JSN and pain progression, whereas adding meniscal scores raised the AUC to 0.722 [9]. Similarly, combining MRI and radiographic markers resulted in an AUC of 0.718, which increased to 0.722 with the addition of biochemical biomarkers [8]. Using urine CTX-II as a predictor only achieved an AUC of 0.583, but combining it with other biochemical biomarkers, clinical variables, and radiographic scores raised the AUC to 0.668 [12]. In our study, using single-structure MRI radiomic features alone yielded AUCs of 0.507 (95% CI [0.424, 0.589]) 0.747 (95% CI [0.685, 0.801]) for predicting JSN and/or pain progression. However, integrating load-bearing tissue of MRI radiomics, the predictive performance of LTB-RM in predicting KOA progression was enhanced compared with single-structure MRI radiomic models. To predict JSN and pain progression, by incorporating biochemical biomarkers, the AUCs (95% CI) of LBT-RM increased from 0.808 (95% CI [0.776, 0.836]) to 0.860 (95% CI [0.832, 0.883]) in the total test cohort. The 6.9% increase in AUC was modest and not statistically significant and therefore must be interpreted cautiously. Furthermore, combining biochemical biomarkers with clinical KOA high-risk factors, the BCM showed similar performance with LBT-RM in predicting KOA progression. Finally, integrating load-bearing tissue of MRI radiomics, biochemical markers, and clinical KOA high-risk factors, the performance of LBTRBC-M improved in predicting KOA progression compared with LBT-RM, with AUCs of 5.7–15.8% increasing and kappa values of 32.5–55.5% increasing in the test cohorts. In addition, LBTRBC-M also outperformed other models for predicting KOA progression, including LBT-MOM, BCM, and LBTMBC-M.

In clinical practice, physicians have limited experience in predicting KOA progression. In our research, using the MOAKS scoring data, assessed by two experienced musculoskeletal radiologists (more than 10 years of experience in assessing KOA radiology) [9], LBT-MOM yielded AUCs ranged from 0.643 (95% CI [0.541, 0.733]) to 0.736 (95% CI [0.668, 0.794]) and accuracy ranged from 48.6% to 52.9% in the test cohorts. The accuracy of resident physicians in our study to predict KOA progression ranged from 44.8% to 48.4% in the test cohorts, which showed similar predictive performance with single-structure MOAKS models, including FE-RM, TI-RM, and MM-RM. However, when provided with the support of LBTRBC-M, resident physicians showed improved accuracy in predicting KOA progression, with accuracy increasing from 46.9% to 65.4% in the total test cohort. In addition, the sensitivity and specificity of physicians to predict JSN and pain progression were also improved in the total test cohort, with the mean sensitivity and specificity increased from 57.5% and 51.8% to 68.1% and 80.4%, respectively. The LBTRBC-M model offers valuable support for resident physicians in predicting KOA progression, improving diagnostic accuracy and aiding clinical decision-making. However, real-world implementation faces key barriers, including high costs for data acquisition and software integration, time constraints for data processing, and the need for clinician training to ensure proper interpretation of results. Additionally, addressing regulatory compliance and ensuring data privacy are essential for successful adoption. Overcoming these challenges through streamlined workflows and institutional support is critical to unlocking the model's full potential in clinical practice.

To mitigate class imbalance, we applied a stratified cohort split in the LBTRBC-M model to ensure proportional representation of each KOA progression subtype. This improved evaluation fairness and revealed that stratification notably

reduced AUC inflation in JSN progression and non-progression groups. These results underscore the necessity of stratified validation in multi-class modeling to enhance generalizability and avoid performance overestimation. In this study, we took several measures to mitigate overfitting in the XGBOOST model. First, we used 10-fold cross-validation to robustly evaluate the model's generalizability. Hyperparameters were optimized via grid search with cross-validation, balancing complexity and performance. L1 and L2 regularization ($\alpha$ and $\lambda$) were applied to reduce noise fitting. Simpler models, such as Logistic Regression and Decision Trees, were also tested to confirm the improvements were meaningful. These steps ensured accurate, generalizable predictions for KOA progression.

Our predictive model represents a step toward developing tools that could support clinical decision-making for KOA progression, pending further validation in diverse populations and clinical settings. It enables personalized risk assessments, aiding clinicians in early interventions and tailored treatment plans to slow disease progression and reduce invasive procedures. The model also supports patient monitoring for timely care adjustments. While further validation in diverse cohorts is needed, it represents a promising tool for enhancing precision medicine and optimizing resource allocation in KOA management.

In this study, we identified key sources of uncertainty, including parameter uncertainty, data uncertainty, and model structure uncertainty. Parameter uncertainty stems from the variability in hyperparameters, such as max_depth, learning_rate, and $\lambda$, which were optimized through grid search and cross-validation. Data uncertainty arises from potential measurement errors in MRI scans, biomarkers, and clinical data, as well as class imbalances and missing data. Model structure uncertainty relates to assumptions made by the XGBOOST algorithm, which may not capture all complex relationships, particularly between biomarkers and imaging features. While we addressed some of these uncertainties through optimization and validation, others, like data collection variations, remain unquantified. Future studies will focus on quantifying these uncertainties and improving the model's robustness.

There are a few limitations to the present study. First, the MRI sequence (SAG-3D-DESS-WE) we used is not commonly employed in clinical practice, so future studies should include multiple common clinical MRI sequences such as Turbo Spin Echo (TSE). Second, our results need validation in independent populations, specifically, we plan to test the model on independent datasets from other cohorts or institutions, such as Multicenter Osteoarthritis Study (MOST) cohort, which would provide a robust assessment of its generalizability across different populations and settings. Third, our study has not incorporated MRI radiomics of all knee joint structures for developing predictive models. In future research, we plan to include more structures, such as the patella, ligaments, synovium, and muscles. Fourth, while the CNN automatic segmentation model we used is suitable for small sample sizes, it is not the latest deep learning architecture. We aim to introduce updated architectures, like the transformer architecture, to enhance segmentation accuracy. Fifth, the MRI radiomics and machine learning algorithms we use are conducive to model interpretation and understanding by clinicians, but their automation is not high. Future efforts will focus on using deep learning algorithms to establish predictive models and improve automation. Sixth, the impact of outliers on model performance was not assessed, but future studies will conduct robustness checks by excluding or mitigating outliers to evaluate their influence. Seventh, while this study focused on XGBOOST, future work will explore additional methods like Generalized Additive models (GAMs) or spline regression to confirm the stability of findings. Eighth, model stability across demographic subsets (e.g., age, sex) was not analyzed; future studies will test diverse subsets to validate findings across groups. Ninth, We also propose incorporating nested cross-validation in future studies to further enhance methodological rigor. Tenth, Scenario analysis can test model robustness under demographic shifts or new treatments. Future studies will include this to enhance clinical applicability. Eleventh, a limitation of our study is the lack of temporal analysis of feature dynamics over time. Exploring changes in MRI radiomic features, biomarkers, and clinical variables across time points could improve model interpretability and predictive accuracy. Future studies will incorporate temporal analysis to better capture disease progression trajectories. Lastly, the generalizability of our findings to other joints requires further validation.

In conclusion, the integrated model using longitudinal MRI radiomics, biochemical biomarkers, and clinical KOA high-risk factors, improved the prediction performance of KOA progression. LBTRBC-M enhances the accuracy of resident physicians to predict KOA progression and these findings need more validation in future trials.

## Ethical approval

All deidentified patient level clinical data, outcome data, and MRI data in our study were obtained from the OAI, an ongoing, multicenter, prospective cohort study designed to identify biomarkers for KOA. The Health Insurance Portability and Accountability Act (HIPAA) compliant protocol of the OAI study had received institutional review board (IRB) approval. The original OAI study protocol was approved by the institutional review boards of all participating centers, including the coordinating center at the University of California, San Francisco (IRB number: 10-00532).

## Supporting information

**S1 Fig.** Flow chart of FNIH OA Biomarkers Consortium cohort study inclusion. FNIH OA Biomarkers Consortium cohort: Foundation of the NIH OsteoArthritis Biomarkers Consortium cohort, MR: Magnetic Resonance, JSW: Joint Space Width, WOMAC: Western Ontario and McMaster Universities Arthritis Index, KLG: Kellgren-Lawrence Grade, BMI: Body Mass Index, BL: Baseline.
(TIF)

**S2 Fig.** The MRI segmentation scheme in our study. MRI: Magnetic Resonance Image.
(TIF)

**S3 Fig.** The results of 10-fold cross-validation to predict knee osteoarthritis progression and contour plot of predicted label probability under actual labels. The quartile of AUC was shown in **A**. The quartile of AUC in each fold was shown in **B**. Contour plot of predicted label probability under actual labels in final LBTRBC-M was shown in C. The 10-fold cross-validation were repeated 100 interactions. AUC: Area Under receiver operating characteristic Curve. JSN: Joint Space Narrowing, LBTRBC-M: Load-Bearing Tissue Radiomic plus Biochemical biomarker and Clinical variable Model.
(TIF)

**S4 Fig.** Feature selection process by LASSO regression in single-structure MRI radiomic models. Panel **(A) to (F)** show the magnitude of scaled parameter estimates for each model (FE-RM, FC-RM, TI-RM, TC-RM, LM-RM, and MM-RM), which indicate the importance of each MRI radiomic feature in predicting KOA progression. Panels **(A), (B), (C), (D), (E), and (F)** represent the magnitude of scaled parameter estimates for FE-RM, FC-RM, TI-RM, TC-RM, LM-RM, and MM-RM, respectively. Panels **(G) to (L)** present the scaled parameter estimates of the same models using the Akaike Information Criterion (AICc) for feature selection, providing an additional measure of model performance and fit. Panels **(M) to (R)** illustrate the weight of features for each model, showing the relative contribution of each selected feature to the overall predictive power of the model. These results highlight the most influential features in each MRI radiomic model for predicting KOA progression. FE-RM: Femur Radiomic Model, FC-RM: Femoral Cartilage Radiomic Model, TI-RM: Tibia Radiomic Model, TC-RM: Tibial Cartilage Radiomic Model, LM-RM: Lateral Meniscal Radiomic Model, MM-RM: Medial Meniscal Radiomic Model, AICc: Akaike Information Criterion, corrected.
(TIF)

**S5 Fig.** Feature selection process by LASSO regression in the LBT-RM and LBTRBC-M models. Panel **(A) and (B)** show the magnitude of scaled parameter estimates for the LBT-RM and the LBTRBC-M, respectively. Panels **(C) and (D)** represent the AICc-based selection of the most important features for both models, offering an alternative approach to assess the performance and fit of the models. Panels **(E) and (F)** show the weight of features in LBT-RM and LBTRBC-M,

respectively, demonstrating how individual features contribute to the predictive power of each model. These visualizations offer a clearer understanding of the key features selected by LASSO regression and their impact on model performance for predicting KOA progression. LBT-RM: Load-Bearing Tissue Radiomic Model, LBTRBC-M: Load-Bearing Tissue Radiomic plus Biochemical Biomarker and Clinical Variable Model, AICc: Akaike Information Criterion, corrected.
(TIF)

**S6 Fig.** The DESS signal feature maps of load-bearing tissues in different groups. DESS signal intensity maps of femur **(A–D)**, femoral cartilage **(E–H)**, tibia **(I–L)**, tibial cartilage **(M–P)**, lateral meniscus **(Q–T)**, medial meniscus **(U–X)** were developed in four groups. The high values of the femur and tibia were detected in JSN and pain progression, and pain progression group. The high value of femoral cartilage, tibial cartilage, lateral meniscus, and medial meniscus were detected in JSN and pain progression, and JSN progression group. JSN: Joint Space Narrowing, DESS: Double Echo Steady-State.
(TIF)

**S7 Fig.** The confusion matrix results of single-structure MRI radiomic models in the test cohorts. The confusion matrix of FE-RM **(A–D)**, FC-RM **(E–H)**, TI-RM **(I–L)**, TC-RM **(M–P)**, LM-RM **(Q–T)**, MM-RM **(U–X)** in the test cohort 1–3 and the total test cohort. The results of test cohort 1, test cohort 2, test cohort 3, and the total test cohort corresponded to baseline, 1-years follow-up, 2-year follow-up, and encompassed the aforementioned follow-up time points. FE-RM: Femur Radiomic Model, FC-RM: Femoral Cartilage Radiomic Model, TI-RM: Tibia Radiomic Model, TC-RM: Tibial Cartilage Radiomic Model, LM-RM: Lateral Meniscal Radiomic Model, MM-RM: Medial Meniscal Radiomic Model.
(TIF)

**S8 Fig.** The comparations of AUC between FE-RM and FE-MOM in predicting KOA progression. The performance of predicting JSN and pain progression **(A–D)**, JSN progression **(E–H)**, pain progression **(I–L)**, and non progression **(M–P)** in FE-RM and FE-MOM in the test cohort 1–3 and the total test cohort. The results of test cohort 1, test cohort 2, test cohort 3, and the total test cohort corresponded to baseline, 1-years follow-up, 2-year follow-up, and encompassed the aforementioned follow-up time points. FE-RM: Femur Radiomic Model, FE-MOM: Femur MOAKS Model, AUC: Area Under receiver operating characteristic Curve, MOAKS: Magnetic resonance imaging OsteoArthritis Knee Score.
(TIF)

**S9 Fig.** The comparations of AUC between FC-RM and FC-MOM in predicting KOA progression. The performance of predicting JSN and pain progression **(A–D)**, JSN progression **(E–H)**, pain progression **(I–L)**, and non progression **(M–P)** in FC-RM and FC-MOM in the test cohort 1–3 and the total test cohort. The results of test cohort 1, test cohort 2, test cohort 3, and the total test cohort corresponded to baseline, 1-years follow-up, 2-year follow-up, and encompassed the aforementioned follow-up time points. FC-RM: Femoral Cartilage Radiomic Model, FC-MOM: Femoral Cartilage MOAKS Model, AUC: Area Under receiver operating characteristic Curve, MOAKS: Magnetic resonance imaging OsteoArthritis Knee Score.
(TIF)

**S10 Fig.** The comparations of AUC between TI-RM and TI-MOM in predicting KOA progression. The performance of predicting JSN and pain progression **(A–D)**, JSN progression **(E–H)**, pain progression **(I–L)**, and non progression **(M–P)** in TI-RM and TI-MOM in the test cohort 1–3 and the total test cohort. The results of test cohort 1, test cohort 2, test cohort 3, and the total test cohort corresponded to baseline, 1-years follow-up, 2-year follow-up, and encompassed the aforementioned follow-up time points. TI-RM: Tibia Radiomic Model, TI-MOM: Tibia MOAKS model, AUC: Area Under receiver operating characteristic Curve, MOAKS: Magnetic resonance imaging OsteoArthritis Knee Score.
(TIF)

**S11 Fig.** The comparisons of AUC between TC-RM and TC-MOM in predicting KOA progression. The performance of predicting JSN and pain progression **(A–D)**, JSN progression **(E–H)**, pain progression **(I–L)**, and non progression **(M–P)** in TC-RM and TC-MOM in the test cohort 1–3 and the total test cohort. The results of test cohort 1, test cohort 2, test cohort 3, and the total test cohort corresponded to baseline, 1-years follow-up, 2-year follow-up, and encompassed the afore-mentioned follow-up time points. TC-RM: Tibial Cartilage Radiomic Model, TC-MOM: Tibial Cartilage MOAKS model, AUC: Area Under receiver operating characteristic Curve, MOAKS: Magnetic resonance imaging OsteoArthritis Knee Score. (TIF)

**S12 Fig.** The comparisons of AUC between LM-RM and LM-MOM in predicting KOA progression. The performance of predicting JSN and pain progression **(A–D)**, JSN progression **(E–H)**, pain progression **(I–L)**, and non progression **(M–P)** in LM-RM and LM-MOM in the test cohort 1–3 and the total test cohort. The results of test cohort 1, test cohort 2, test cohort 3, and the total test cohort corresponded to baseline, 1-years follow-up, 2-year follow-up, and encompassed the afore-mentioned follow-up time points. LM-RM: Lateral meniscus radiomic model, LM-MOM: Lateral meniscus MOAKS model, AUC: Area Under receiver operating characteristic Curve, MOAKS: Magnetic resonance imaging OsteoArthritis Knee Score. (TIF)

**S13 Fig.** The comparisons of AUC between MM-RM and MM-MOM in predicting KOA progression. The performance of predicting JSN and pain progression **(A–D)**, JSN progression **(E–H)**, pain progression **(I–L)**, and non progression **(M–P)** in MM-RM and MM-MOM in the test cohort 1–3 and the total test cohort. The results of test cohort 1, test cohort 2, test cohort 3, and the total test cohort corresponded to baseline, 1-years follow-up, 2-year follow-up, and encompassed the aforemen-tioned follow-up time points. MM-RM: Medial meniscus radiomic model, MM-MOM: Medial meniscus MOAKS model, AUC: Area Under receiver operating characteristic Curve, MOAKS: Magnetic resonance imaging OsteoArthritis Knee Score. (TIF)

**S14 Fig.** The confusion matrix results of load-bearing tissue MRI radiomic models in the test cohorts. The confusion matrix of LBT-RM **(A–D)** and LBTRBC-M **(E–H)** in the test cohort 1–4 and the total test cohort. The results of test cohort 1, test cohort 2, test cohort 3, and the total test cohort corresponded to baseline, 1-years follow-up, 2-year follow-up, and encompassed the aforementioned follow-up time points. LBT-RM: Load-Bearing Tissue Radiomic Model, LBTRBC-M: Load-Bearing Tissue Radiomic plus Biochemical biomarker and Clinical variable Model. (TIF)

**S15 Fig.** Performance of resident physicians and models in predicting the KOA progression in different time point test cohorts. AUC for predicting JSN and pain progression **(A–F)**, JSN progression **(G–L)**, pain progression **(M–R)**, and non progression **(S–X)** among the LBTRBC-M, LBTMBC-M, and average performance of all resident physicians without (blue dot) and with (red dot) the support of LBTRBC-M in the test cohort 1–3 was demonstrated. As shown in **A, C, E, G, I, K, M, O, Q, S, U, and W**, both the sensitivity and specificity of resident physicians were improved when aid was sup-ported by LBTRBC-M (black arrow) in the test cohort 1–3. As shown in **B, D, F, H, J, L, N, P, R, T, V, and X**, the individual performance of resident physicians was represented by open shapes (without LBTRBC-M aid) and filled shapes (with LBTRBC-M aid). The results of test cohort 1, test cohort 2, and test cohort 3 corresponded to baseline, 1-years follow-up, and 2-year follow-up time points. The colored dotted line of yellow, orange, purple, green, blue, teal, and red represented the predictive performance change of Liu, Zhao, Cao, J Li, Chen, X Wang, Dang, and M Zhang, respectively. AUC: Area Under receiver operating characteristic Curve, LBTRBC-M: Load-Bearing Tissue Radiomic plus Biochemical biomarker and Clinical variable Model, LBTMBC-M: Load-Bearing Tissue MOAKS plus Biochemical biomarker and Clinical variable Model, MOAKS: Magnetic resonance imaging OsteoArthritis Knee Score, AI: Artificial Intelligence. (TIF)

**S16 Fig.** The predictive performance of the LBTRBC-M model using/without using the stratified cross-validation in the total test cohort. **(A–B)** We implemented a stratified cohort split for the LBTRBC-M model to ensure proportional representation of each KOA progression subtype, maintaining an approximate 2:1:1:2 ratio of JSN and pain progression, JSN progression, pain progression, and non-progression. **(C–D)** The AUC of LBTRBC-M model using/without using the stratified cross-validation was displayed in the total test cohort. AUC: Area Under receiver operating characteristic Curve, LBTRBC-M: Load-Bearing Tissue Radiomic plus Biochemical biomarker and Clinical variable Model, JSN: Joint Space Narrowing, KOA: Knee Osteoarthritis.
(TIF)

**S1 Table.** MRI protocol details.
(DOCX)

**S2 Table.** Baseline characteristics of participants in the development cohort 1 and test cohort 1.
(DOCX)

**S3 Table.** Baseline biochemical biomarker levels of participants in the development cohort 1 and test cohort 1.
(DOCX)

**S4 Table.** DSCs for the CNNs automated segmentation and manual adjustment segmentation.
(DOCX)

**S5 Table.** Selected features of predictive models in total development cohort.
(DOCX)

**S6 Table.** The areas under ROC curves of predictive models in the test cohorts.
(DOCX)

**S7 Table.** The accuracy of predictive models in the test cohorts.
(DOCX)

**S8 Table.** Comparing the areas under two correlated ROC curves between predictive models in the test cohorts.
(DOCX)

**S9 Table.** Related risks of outcomes for predictive model outputs.
(DOCX)

**S10 Table.** Predictive performance of resident physicians under the assistance of LBTRBC-M.
(DOCX)

**S11 Table.** The accuracy of resident physicians under the assistance of LBTRBC-M.
(DOCX)

**S12 Table.** Predictive performance of resident physicians under the assistance of LBTRBC-M in the test cohorts.
(DOCX)

**S13 Table.** Related risks of outcomes for LBTRBC-M outputs using different GEE model.
(DOCX)

**S14 Table.** Compares the predictive performance of the LBTRBC-M model using/without using the MI in the total test cohort.
(DOCX)

**S15 Table.** Comparing predictive performance of the LBTRBC-M model using different algorithms in the total test cohort. (DOCX)

**S16 Table.** The parameters of LBTRBA-M model. (DOCX)

**S17 Table.** The predictive performance of LBTRBC-M using different hyperparameters in the total test cohort. (DOCX)

**S18 Table.** Compares the predictive performance of the LBTRBC-M model with different interactions in the total test cohort. (DOCX)

**S19 Table.** Compares the predictive performance of the LBTRBC-M model using/without using the stratified cross-validation in the total test cohort. (DOCX)

**S1 Checklist.** TRIPODAI checklist. (PDF)

## Acknowledgments

We would like to acknowledge the dedication and commitment of the OAI study participants. The OAI is a public-private partnership comprised of five contracts (N01-AR-2-2258; N01-AR-2-2259; N01-AR-2-2260; N01-AR-2-2261; N01-AR-2-2262) funded by the NIH and conducted by the OAI Study Investigators. Private funding partners include Merck Research Laboratories, Novartis Pharmaceuticals Corporation, GlaxoSmithKline, and Pfizer, Private sector funding for the OAI is managed by the foundation for the NIH. This manuscript was prepared using an OAI public use data set (in addition to data obtained within NIH/NIAMS funded ancillary grants) and does not necessarily reflect the opinions or views of the OAI investigators, the NIH, or the private funding partners. Special thanks go to the subjects who made this study possible, the OAI investigators, the Foundation of the NIH Osteoarthritis Biomarkers Consortium investigators, staff, and participants. We used ChatGPT for language editing in this study.

## Author contributions

**Conceptualization:** Tianyu Chen, Changhai Ding, Shengfa Li.

**Data curation:** Ting Wang, Changhai Ding, Shengfa Li.

**Formal analysis:** Peihua Cao, Jia Li, Xiaoshuai Wang, Shengfa Li.

**Funding acquisition:** Changhai Ding, Shengfa Li.

**Investigation:** Qin Dang, Shengfa Li.

**Methodology:** Ting Wang, Hao Liu, Wenbo Zhao, Peihua Cao, Jia Li, Tianyu Chen, Guangfeng Ruan, Yan Zhang, Xiaoshuai Wang, Qin Dang, Mengdi Zhang, Alexander Tack, Shengfa Li.

**Project administration:** Shengfa Li.

**Resources:** Shengfa Li.

**Software:** Tianyu Chen, Guangfeng Ruan, Mengdi Zhang, Alexander Tack, Shengfa Li.

**Supervision:** Changhai Ding, Shengfa Li.

**Validation:** Yan Zhang, Shengfa Li.

**Visualization:** Shengfa Li.

**Writing – original draft:** Ting Wang.

**Writing – review & editing:** David Hunter, Changhai Ding, Shengfa Li.

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
