## [Editor Report · Decision Letter 0]

Dear Dr Li, 

Thank you for submitting your manuscript entitled "Predicting KOA Progression: Integrating Neural network, Longitudinal MRI Radiomics, and Biochemical Biomarkers" for consideration by PLOS Medicine.

Your manuscript has now been evaluated by the PLOS Medicine editorial staff and I am writing to let you know that we would like to send your submission out for external peer review.

Please re-submit your manuscript within two working days, i.e. by Aug 22 2024.

Feel free to email me at atosun@plos.org or us at plosmedicine@plos.org if you have any queries relating to your submission.

Kind regards,

Alexandra Tosun, PhD

Associate Editor

PLOS Medicine

---

## [Decision Letter · Decision Letter 1]

Dear Dr Li,

Many thanks for submitting your manuscript "Predicting KOA Progression: Integrating Neural network, Longitudinal MRI Radiomics, and Biochemical Biomarkers" (PMEDICINE-D-24-02732R1) to PLOS Medicine. The paper has been reviewed by subject experts and a statistician; their comments are included below and can also be accessed here: [LINK]

As you will see, the reviewers find the study interesting, but point out the lack of detail and the need to clarify the methodology. After discussing the paper with the editorial team and an academic editor with relevant expertise, I'm pleased to invite you to revise the paper in response to the reviewers' comments. We plan to send the revised paper to some or all of the original reviewers, and we cannot provide any guarantees at this stage regarding publication.

We ask that you submit your revision by Dec 19 2024. However, if this deadline is not feasible, please contact me by email, and we can discuss a suitable alternative.

Don't hesitate to contact me directly with any questions (atosun@plos.org). 

Best regards, 

Alexandra 

Alexandra Tosun, PhD 

Associate Editor

PLOS Medicine

atosun@plos.org

Comments from the reviewers: 

Reviewer #1: 1. The use of various statistical tests (e.g., unpaired t-test, one-way ANOVA, χ² test, Mann-Whitney test, Kruskal-Wallis test) for comparing different groups is appropriate. However, the rationale behind selecting specific tests for continuous versus categorical variables should be explicitly stated. Clarifying why a parametric or non-parametric test was chosen, based on the data distribution, would enhance the transparency of the methodology.

2. The use of Generalized Estimating Equations (GEE) for analyzing correlated data, such as repeated measures, is appropriate. However, more details should be provided on the correlation structure assumed in the GEE model. Additionally, it would be beneficial to specify how the model handles missing data and whether any sensitivity analyses were conducted to assess the robustness of the findings.

3. The application of LASSO for feature selection is valid, but further details could be provided on how the optimal penalty parameter (lambda) was chosen.

4. 10-fold cross-validation method used to validate model performance is appropriate. However, the manuscript should describe whether stratified cross-validation was applied, particularly when dealing with imbalanced datasets, as this could significantly impact the performance metrics.

5. Using LASSO regression and cross-validation for both feature selection and model validation carries a risk of overfitting, especially if the same data is used for both processes. The authors should clarify how they mitigated feature selection bias, potentially through techniques like nested cross-validation or using a separate validation set.

6. The manuscript mentions multiple tests, but it lacks a clear indication of how multiple comparisons were managed to control for Type I error. If multiple comparisons were indeed made, methods such as Bonferroni correction or false discovery rate (FDR) control should be discussed to ensure the robustness of the findings.

7. The ROC analysis and DeLong test are standard methods for comparing the performance of predictive models. The authors should ensure that the interpretation of AUC, sensitivity, specificity, and kappa values is clear and tied to the clinical relevance of the findings. Including confidence intervals for the AUC values would also provide a sense of the precision of these estimates.

8. It is essential to check and report whether the assumptions underlying each statistical model were verified. For example, for LASSO regression, assumptions related to the distribution of errors and the linearity of relationships should be considered. Similarly, for GEE models, the choice of correlation structure (e.g., exchangeable, autoregressive) should be justified based on the data characteristics.

9. While cross-validation was employed, it would be beneficial to discuss any plans or considerations for external validation. If external datasets are available, they should be used to validate the model's performance, as external validation is crucial for assessing the generalizability of predictive models.

10. While the manuscript briefly mentions the handling of missing data, a more detailed discussion is warranted. The authors should specify how missing data were handled (e.g., imputation methods, exclusion) and whether any sensitivity analyses were performed to assess the impact of missing data on the study's findings.

11. The methods section should discuss how potential confounders were addressed in the statistical models. For instance, were any variables included to adjust for confounding? If so, how were these variables selected?

12. The manuscript does not mention whether interaction terms were considered in the models. Given the complexity of the relationships between radiomic features, biomarkers, and clinical variables, it might be worthwhile to explore interaction terms to capture any synergistic effects. If interactions were tested but found not to be significant, this should be clearly stated.

13. If there are subgroups within the data (e.g., based on demographic variables like age or sex), the authors might consider stratified analysis or clustering techniques to explore whether the model performs differently across these subgroups. 

14. While LASSO and XGBOOST can handle some non-linearity, it might be worth exploring whether more explicitly non-linear modeling approaches (e.g., spline regression, generalized additive models) could provide better fits for certain variables, especially those that exhibit complex relationships with the outcome.

15. Given that XGBOOST and LASSO involve the selection of hyperparameters, the manuscript should discuss how sensitive the results are to different hyperparameter choices. Grid search or Bayesian optimization methods for hyperparameter tuning could be mentioned to ensure that the model is optimally configured.

16. While XGBOOST is a powerful algorithm, the authors might consider discussing why this specific method was chosen over other machine learning techniques (e.g., Random Forest, Support Vector Machines, Neural Networks). A brief comparison or justification could strengthen the manuscript's methodological rigor.

17. While the manuscript discusses techniques like cross-validation, it would be prudent to further elaborate on how the risk of overfitting was mitigated, particularly given the complexity of models like XGBOOST. The discussion could include the importance of balancing model complexity with simplicity in design.

18. Authors could include robustness checks to test the stability of their findings under different assumptions or subsets of data. For example, re-running analyses after excluding outliers, using different modeling techniques, or testing the model on various subsets of data could provide additional confidence in the results.

19. Given the complexity of the models, the authors should discuss how uncertainty in the model predictions is quantified and reported. This could include confidence intervals, prediction intervals, or Bayesian approaches that provide a probabilistic interpretation of model outputs.

20. The manuscript could benefit from a discussion on scenario analysis, where the model's predictions are tested under different hypothetical scenarios. This approach could help in understanding how changes in key variables (e.g., a sudden change in patient demographics or the introduction of a new treatment) might impact predictions.

21. While the manuscript compares the proposed model with other models, a more detailed comparative analysis with traditional statistical methods or simpler machine learning models (e.g., logistic regression, decision trees) could provide insights into the added value of the more complex modeling approaches used. 

22. The manuscript discusses sensitivity and specificity, but it would be valuable to delve deeper into the trade-offs between these metrics. For instance, depending on the clinical application, different thresholds might be appropriate, and ROC curves or precision-recall curves could be used to explore these trade-offs in more detail.

23. The statistical analysis should be carefully interpreted within the context of clinical significance, not just statistical significance. The authors should avoid over-reliance on p-values and instead emphasize effect sizes and confidence intervals.

Additional comments: 

24. The statistical analyses were performed using R (version 4.1.1) and SAS (version 9.4). The authors should ensure that all relevant packages (e.g., pROC in R) and their versions are cited. This is crucial for ensuring the reproducibility of the analyses.

25. It is commendable that the authors have made their data and source code publicly available. However, they should also ensure that all the scripts used for the analysis are well-documented and that a clear workflow is provided. This would facilitate others in reproducing the results.

26. Ensure that all figures and their legends in the supplementary materials are self-explanatory. For example, Figure S4 and Figure S5 detail the feature selection process using LASSO regression, but the legends could benefit from a more detailed explanation of the significance of the selected features and their impact on model performance. Adding more context would help readers who may not be intimately familiar with LASSO or radiomic features.

27. Supplementary tables, such as Table S2 and Table S3, contain extensive data on baseline characteristics and biomarker levels. Each table should have clear annotations explaining what each biomarker or clinical variable represents, especially for an interdisciplinary audience. For instance, terms like "sCOMP" or "sHA" should be briefly described or referenced to where a description can be found.

28. Including a workflow diagram in the supplementary materials could aid in understanding complex processes. For example, a diagram showing the steps from raw MRI data through to the final model could be beneficial, especially in understanding the MRI segmentation scheme outlined in Figure S2.

Reviewer #2: Osteoarthritis (OA) is the most frequent musculoskeletal disease and the knee OA is the most frequent symptomatic site involved.

In this paper authors aimed to provide a new model of KOA progression prediction using both clinical, biochemical and imaging parameters based on the FNIH biomarker consortium.

This paper is relevant and contributive to the field however there are some issues which must be revised before the paper can be accepted.

Major issues:

1. It is quite unusual to see modeling processes with 25,000 repetitions for cross-validation. Typically, standard practices involve around 100 iterations, with 300 iterations being a common upper limit unless the data complexity demands more. This excessive number of iterations might lead to diminishing returns and could indicate overfitting if not carefully monitored. It therefore would be beneficial for the authors to clarify the rationale behind such a high number of repetitions. They should explain how this approach improves predictive performance and whether they observed any significant differences in outcomes compared to more conventional methods. This would help strengthen their methodology section and provide a clearer justification for their choices.

2. The extensive iterations used in both the cross-validation and feature selection processes lend credibility to the predictive modeling efforts in the study. However, it is crucial to address the computational feasibility and interpret the potential implications of such high iteration counts. Overall, these aspects contribute to the manuscript's strength and could be emphasized further to showcase the rigor of the research methodology.

3. Clarification on Feature Selection Process: The methodology mentions using LASSO for feature selection before applying the XGBOOST algorithm. Providing more details on the feature selection process, including criteria for inclusion or exclusion of features, would improve transparency and reproducibility.

4. Temporal Analysis and Model Interpretation: Given the longitudinal nature of the data, discussing how changes over time influence the predictive model could be beneficial. Analyzing the temporal dynamics of specific features and their relationship with KOA progression would provide deeper insights into the disease's progression and the model's interpretability.

5. Consideration of Clinical Relevance: Discussing the clinical implications of their predictive model, including how it could be applied in real-world settings and its potential impact on patient management, would add significant value to the manuscript.

6. A more comprehensive review of the literature, citing prominent groups and contextualizing this study among both classic and recent research, would enhance the scientific rigor and relevance of the manuscript.

The lack of citations from influential groups such as Saarakkala or Lespessailles raises concerns about comprehensiveness in the literature review.

Minor issues:

1. P7, second paragraph: "The main analysis focused on comparing knees with both JSN

and pain progression to all other knees. (16)". I wonder if ref (16) is appropriate because it is an old reference (2014) and a position paper rather than an original article.

2. P8, again, (ref18) reported the effects of sprifermin is a clinical trial and thus is not appropriate.

3. P9, ref (19) is the main reference to describe the KL method but do not explain in the present study how and who analyses the radiographs.

4. P18, BCM, this has not been defined previously.

5. P19, in fact, a recent paper has investigated the combination of biomechanical and radiological biomarkers to predict KOA progression in the FNIH dataset (https://doi.org/10.3390/biomedicines12030666 Performance of Radiological and Biochemical Biomarkers in Predicting Radio-Symptomatic Knee Osteoarthritis Progression- Biomedicines-2024).

6. P19, authors wrote « Similarly, combining MRI and radiographic markers resulted in an AUC of 0.718, which increased to 0.722 with the addition of biochemical biomarkers.(8) ». However moving from 0.718 to 0.722, was this numerically better AUC statistically and clinically significant?

Reviewer #3: In this paper, the authors propose a diagnostic and prognostic model for knee osteoarthritis progression. Using data from the Osteoarthritis Initiative (OAI) and advanced modeling techniques like XGBoost, convolutional neural networks, and longitudinal MRI radiomics, the study demonstrates the potential to improve prediction accuracy for joint space narrowing and pain progression. While this work is innovative and aligns with the journal's scope, significant revisions are necessary before publication.

-The introduction lacks a clear articulation of the study's specific goals and the gaps it aims to address.

-While the background is comprehensive, it could be condensed to focus more on relevant prior studies and their shortcomings.

-Please provide more details on the statistical and machine learning methodologies, such as the rationale behind using LASSO logistic regression for feature selection and its effect on model interpretability

-Please, explain why a 1:1 split between the development and test cohorts was used, and discuss how this split ensures robust external validation. How does this approach to cohort splitting compare to alternative validation strategies, such as k-fold cross-validation, in terms of generalizability?

-While the results are robust and well-documented, the authors should clarify : What defines a "successful" prediction? (e.g., accuracy threshold or clinical utility).

- In the Discussion Section please include a more explicit comparison with recent literature, highlighting where this study excels or falls short.

- While the discussion mentions the potential utility of the model for resident physicians, more emphasis is needed on real-world implementation barriers (e.g., cost, time, and training).

- In some sections the English language needs to be polished to improve fluency and clarity

---

* Please upload any figures associated with your paper as individual TIF or EPS files with 300dpi resolution at resubmission; please read our figure guidelines for more information on our requirements: http://journals.plos.org/plosmedicine/s/figures. While revising your submission, please upload your figure files to the PACE digital diagnostic tool, https://pacev2.apexcovantage.com/. PACE helps ensure that figures meet PLOS requirements. To use PACE, you must first register as a user. Then, login and navigate to the UPLOAD tab, where you will find detailed instructions on how to use the tool. If you encounter any issues or have any questions when using PACE, please email us at PLOSMedicine@plos.org.

* FINANCIAL DISCLOSURES: The funding statement should include: specific grant numbers, initials of authors who received each award, URLs to sponsors’ websites. Also, please state whether any sponsors or funders (other than the named authors) played any role in study design, data collection and analysis, the decision to publish, or preparation of the manuscript. If they had no role in the research, include this sentence: “The funders had no role in study design, data collection and analysis, decision to publish, or preparation of the manuscript.”

* DATA AVAILABILITY: The Data Availability Statement (DAS) requires revision. For each data source used in your study:

* ETHICS STATEMENT: Please provide the name(s) of the institutional review board(s) that provided ethical approval. Please specify whether informed consent was written or oral.

FIGURES AND TABLES

SUPPLEMENTARY MATERIAL

REFERENCES

* Where website addresses are cited, please include the complete URL and specify the date of access (e.g. [accessed: 12/06/2024]).

STUDY TYPE-SPECIFIC REQUESTS 

The following list is derived from Geoffrey P Garnett, Simon Cousens, Timothy B Hallett, Richard Steketee, Neff Walker. Mathematical models in the evaluation of health programmes. (2011) Lancet DOI:10.1016/S0140-6736(10)61505-X: 

* If pertinent, please provide a diagram that shows the model structure, including how the natural history of the disease is represented, the process and determinants of disease acquisition, and how the putative intervention could affect the system.

* Please provide a complete list of model parameters, including clear and precise descriptions of the meaning of each parameter, together with the values or ranges for each, with justification or the primary source cited and important caveats about the use of these values noted.

* Please provide a clear statement about how the model was fitted to the data, including goodness-of-fit measure, the numerical algorithm used, which parameter varied, constraints imposed on parameter values, and starting conditions.

* For uncertainty analyses, please state the sources of uncertainties quantified and not quantified [can include parameter, data, and model structure].

* Please provide sensitivity analyses to identify which parameter values are most important in the model. Uncertainty estimates seek to derive a range of credible results on the basis of an exploration of the range of reasonable parameter values. The choice of method should be presented and justified.

* Please discuss the scientific rationale for the choice of model structure and identify points where this choice could influence conclusions drawn. Please also describe the strength of the scientific basis underlying the key model assumptions.

* For studies that develop a prediction model or evaluate its performance, please ensure that the study is reported according to the TRIPOD statement (https://www.equator-network.org/reporting-guidelines/tripod-statement) and include the completed checklist as Supporting Information. Please add the following statement, or similar, to the Methods: "This study is reported as per the Transparent Reporting of a Multivariable Prediction Model for Individual Prognosis Or Diagnosis (TRIPOD) statement (S1 Checklist)." For studies using machine learning, please use the TRIPOD-AI checklist. When completing the checklist, please use section and paragraph numbers, rather than page numbers.

---

## [Decision Letter · Decision Letter 2]

Dear Dr Li,

Many thanks for re-submitting your manuscript "Predicting KOA Progression: Integrating Neural network, Longitudinal MRI Radiomics, and Biochemical Biomarkers" (PMEDICINE-D-24-02732R2) to PLOS Medicine. The paper has been seen again by one subject expert and the statistician; their comments are included below and can also be accessed here: [LINK]

Thank you for your response to the reviewers' comments. As you can see, while the statistical reviewer is satisfied with your responses to their comments, the subject-matter reviewer still has concerns about your study. Please note that any changes you make in response to the reviewer and/or editorial comments must be within the text and provide full clarity to the reviewers for us to consider the manuscript further. It's not sufficient (in all cases) to address the reviewer comments by adding them as a limitation. After discussing the paper with the editorial team, we ask you to carefully and robustly address the comments in a further revision. We plan to send the revised paper to some or all of the original reviewers.

We ask that you submit your revision by Mar 07 2025. However, if this deadline is not feasible, please contact me by email, and we can discuss a suitable alternative.

Don't hesitate to contact me directly with any questions (atosun@plos.org). 

Best regards, 

Alexandra 

Alexandra Tosun, PhD 

Associate Editor

PLOS Medicine

atosun@plos.org

Comments from the reviewers: 

Reviewer #1: Thank you for thoroughly addressing all my comments and providing detailed clarifications in your revised manuscript. I appreciate the effort you have put into enhancing the rigor and transparency of your analyses.

The manuscript now reflects a robust and valuable contribution to subject, I am happy to recommend this work for publication and look forward to seeing its positive impact in the field.

Reviewer #2: R1- We thank the authors for clarifying their rationale for performing 25,000 repetitions of cross-validation. While ensuring stability in performance estimates is important, the justification for such an unusually high number remains unclear, especially since they acknowledge that predictive accuracy plateaued before reaching this level. Beyond that point, the additional computational burden becomes difficult to justify. A quantitative analysis demonstrating how variance decreases across iterations would strengthen their argument. Their plan to explore fewer iterations in future studies is good, but a more systematic justification for their current approach is essential to enhance the study's methodological transparency and rigor. So we request the authors to include this justification in the current manuscript.

R2- Once again, we thank the authors for recognizing the significance of clarifying the implications of their high number of iterations, especially with relation to computing feasibility. Although they use robustness to support their decision, further explanation is required to properly address methodological trade-offs and feasibility.

The computational feasibility is still unclear. Cross-validation with 25,000 iterations can be computationally challenging, particularly when dealing with high-dimensional data. To show that their method is practical and scalable, it would be beneficial if the authors provide information on hardware specs, runtime, and resource needs.

The authors also do not provide quantitative proof to support the need for 25,000 iterations, even though they claim that less iterations (such as 100-300) resulted in higher variability in performance indicators. A statistical analysis (such as a graphic showing the relationship between variability) would show clearly where performance metrics stabilize and whether this high number of iterations is justified.

R3- We appreciate the authors' detailed clarification of the LASSO feature selection process, which greatly enhances the clarity of their methodology. As an optional improvement, we suggest including a comparison between the number of features initially considered and those retained after LASSO selection.

R4- Authors response acknowledges the limitation regarding the lack of temporal analysis of feature dynamics over time. Addressing it as a limitation in their discussion section is sufficient.

R5- We appreciate the authors' detailed response concerning the clinical relevance of their predictive model. 

R6- Although the authors have cited 3 recent papers from 3 different research teams, the conclusions drawn about their results are not entirely fair and balanced. Specifically, they state, 'Compared to these studies, our model (LBTRBC-M) achieved higher AUC values (0.869-0.920) by integrating load-bearing tissue MRI radiomics, biochemical biomarkers, and clinical variables. This suggests a more comprehensive risk assessment and superior predictive accuracy.' However, the results from their study and those from the other cited studies cannot be directly compared, as the datasets and methodologies used in those studies differ. Comparing AUC values is only relevant in the context of 'challenge' studies, where different teams use the same dataset and aim for the same goal, as seen in the KOA progression prediction challenge (KNOAP2020, Ref.1). Therefore, the sentence from Line 526 to Line 529 should be deleted.

Ref.1: Hirvasniemi et al., The KNee OsteoArthritis Prediction (KNOAP2020) challenge: An image analysis challenge to predict incident symptomatic radiographic knee osteoarthritis from MRI and X-ray images, Osteoarthritis Cartilage (2023), 31(1):115-125. doi:10.1016/j.joca.2022.10.001 

Furthermore, I have several additional concerns that the authors should address before I can give positive consideration for publication: 

Major concerns:

1) Although authors indicate that both TUKEY'S honestly significant difference and Bonferroni correction were applied to control for type I error due to multiple comparisons, they should provide to the reader the nominal P value which is now considered as statistically significant in their work. Thus, the P value indicated line 375 in the statistical analysis section is probably wrong due to the numerous comparisons made in the present study. 

2) While DeLong's test is widely recognized as a non-parametric method for comparing the AUCs of different models, it does not fully account for the shared variance between nested models, as in this study. This limitation can lead to biased or overly conservative results (Demler et al., 2012; Ref. 1). To address this issue, the authors should consider using alternative approaches, such as bootstrap resampling or permutation-based testing.

Ref.1: Demler et al., Misuse of DeLong test to compare AUCs for nested models. Statistics in medicine, 31.23 (2012): 2577-2587

3) Line 563-566: Typically, the clinical relevance of results is discussed when a statistically significant difference is observed. However, if the effect size is limited, the clinical relevance may be considered poor. In this study, the authors take a different approach, suggesting that a non-statistically significant result may still hold clinical relevance. I find this reasoning inappropriate, as statistical significance is a fundamental criterion for establishing meaningful clinical interpretation. Moreover, if the authors wish to argue for clinical relevance despite the absence of statistical significance, they should provide additional supporting evidence through other clinical metrics or relevant justifications, like PPV and NPV.

4) Lines 327-329: The authors chose a 25% threshold for displaying results in red font based on the output of the LBTRBC-M model. However, this threshold seems somewhat arbitrary without further justification. It would be beneficial for the authors to provide a rationale for selecting 25% as the cut-off point, as this decision could impact on the interpretation of the results.

5) Line 599: The authors did not implement stratified cross-validation, which significantly affects the validity of their comparisons to other studies. Without stratification, the distribution of cases and controls across folds may be highly imbalanced, leading to misleading model performance metrics. In particular, folds with a much lower Case/Control ratio can artificially inflate the AUC, as the model may achieve high accuracy primarily by correctly predicting the majority class (controls) rather than effectively distinguishing progressors. This can create an overestimation of the model's discriminatory power and reduce the reliability of the results. To ensure a fair and meaningful comparison with other studies and to obtain robust performance estimates, the authors should implement stratified cross-validation before final acceptance. This will help maintain a consistent class distribution across folds, preventing potential bias in model evaluation. Acknowledging the lack of stratification in the cross validation process as the authors do in their revised manuscript is not sufficient for quality of this kind of study. 

Minor concerns:

1) Throughout the paper, the authors should consistently specify that they are using "MRI radiomics". When 'radiomics' is mentioned without this clarification, readers may assume that X-ray radiomics was also studied, which is not the case. 

2) The authors do not report the 95% confidence interval (CI) values for the AUC throughout the paper. Reporting the 95% CI is essential for assessing the precision and reliability of the AUC estimates. Without these intervals, it is difficult to evaluate the statistical significance and the robustness of the model's performance. I strongly recommend including the 95% CI for AUC to provide a more comprehensive interpretation of the results.

3) Line 154: The period of KOA progression should be specified upon its first mention. For example, it should be stated as 'KOA progression over 4 years' or 'KOA progression within the subsequent 2 years.

4) Line 171: It would be better to specify that the 194 cases refer to those related to JSN and pain cases.

5) Line 217: It would be necessary to specify which serum and/or urine biochemical markers were included or at least provide a reference. The authors mention the number of biochemical markers on line 322, but do not specify which markers were studied, and this information is not provided until after four pages.

6) Lines 410, 508, 510 and 559: It is important to specify what type of progression this value refers.

---

## [Decision Letter · Decision Letter 3]

Dear Dr. Li,

Thank you very much for re-submitting your manuscript "Predicting KOA Progression: Integrating Neural network, Longitudinal MRI Radiomics, and Biochemical Biomarkers" (PMEDICINE-D-24-02732R3) for review by PLOS Medicine.

Thank you for your detailed response to the reviewers' and editors’ comments. I have discussed the paper with my colleagues, and it has also been seen again by two of the original reviewers. The changes made to the paper were satisfactory to the reviewers. As such, we intend to accept the paper for publication, pending your attention to the reviewers' and editors' comments below in a further revision. When submitting your revised paper, please once again include a detailed point-by-point response to the editorial comments.

[LINK]

In revising the manuscript for further consideration here, please ensure you address the specific points made by each reviewer and the editors. In your rebuttal letter you should indicate your response to the reviewers' and editors' comments and the changes you have made in the manuscript. Please submit a clean version of the paper as the main article file. A version with changes marked must also be uploaded as a marked up manuscript file. Please also check the guidelines for revised papers at http://journals.plos.org/plosmedicine/s/revising-your-manuscript for any that apply to your paper.

We ask that you submit your revision within 1 week (Jun 18 2025). However, if this deadline is not feasible, please contact me by email, and we can discuss a suitable alternative.

Please do not hesitate to contact me directly with any questions (atosun@plos.org). If you reply directly to this message, please be sure to 'Reply All' so your message comes directly to my inbox.

We look forward to receiving the revised manuscript.   

Sincerely,

Alexandra Tosun, PhD

Senior Editor 

PLOS Medicine

plosmedicine.org

Comments from Reviewers:

Reviewer #2: 

I would like to sincerely thank the author to have consistently addressed all my concerns or comment in their last revised manuscript.

I am glad to consider that with its revisions the paper is now recommendable for publication.

[LINK]

Requests from Editors:

GENERAL

* Please confirm that your title complies with to PLOS Medicine's style. Your title must be nondeclarative and not a question. It should begin with main concept if possible. "Effect of" should be used only if causality can be inferred, i.e., for an RCT. Please place the study design ("A randomized controlled trial," "A retrospective study," "A modelling study," etc.) in the subtitle (i.e., after a colon).

* Statistical reporting: Please revise throughout the manuscript, including tables and figures.

- Please report statistical information as follows to improve clarity for the reader ""22% (95% CI [13,28]; p</=)"".

- Please separate upper and lower bounds with commas instead of hyphens as the latter can be confused with reporting of negative values.

- Please repeat statistical definitions (HR, CI etc.) for each set of parentheses.

* Please ensure that all abbreviations are defined at first use throughout the text (including statistical abbreviations). Please also check figures and tables.

* Please ensure that tables and figures, including those in supplementary files, are appropriately referenced in the main text.

* Please note that the funding statement should also include URLs to sponsors’ websites.

* Data availability: Please clarify whether, once access to the repository is gained (via the link you provided), it will be clear to anyone interested in the data which dataset was used for this study.

* Many abbreviations are used throughout the main text. Please check whether the number could potentially be reduced to improve readability. Please also carefully check whether abbreviations are defined at first use throughout the text (including statistical abbreviations). Please also check figures and tables.

* Please consider whether removing some of the numerical results in main text would improve readability in cases where the numbers are easy to find in the relevant table/figure.

* The manuscript is still quite complex and, at times, difficult to follow, particularly considering that PLOS Medicine serves a broad medical audience. Please keep this in mind when revising the manuscript. Please streamline the manuscript and create easy-to-follow methods and results sections. These sections should enable readers who are unfamiliar with the topic to understand your research.

* Because Github depositions can be readily changed or deleted, we encourage you to make a permanent DOI'd copy in Zenodo and provide the URL.

ABSTRACT

* Please confirm that your abstract complies with our requirements, including providing all the information relevant to this study type https://journals.plos.org/plosmedicine/s/submission-guidelines#loc-abstract

* Please ensure that all numbers presented in the abstract are present and identical to numbers presented in the main manuscript text.

* Please suggest changing the description of outcomes to the following: “Outcomes included 1) both Joint Space Narrowing (JSN) and pain progression, 2) only JSN progression, 3) only pain progression, and 4) non-progression (JSN or pain)”

* We don’t think it’s currently 100% clear what you mean with “with a ratio of 2:1:1:2”. Please revise for clarity. We suggest including the exact numbers in the parentheses of the development and the total test cohort.

* “A total of 1753 knee MRIs were included over a 2-year follow-up.” We suggest describing this more accurately, providing the numbers for each time point.

*We suggest adding the definitions of JSN progression and pain progression in the Abstract.

* Please include basic demographics of the participants, i.e. mean age, sex, ethnicity/race.

* Please define MRI at first use.

* Why do you only report the AUC of the test and not the development cohort?

* Please include the number of resident physicians. 

* Abstract Conclusions:

- Please address the study implications without overreaching what can be concluded from the data; the phrase ""In this study, we observed ..."" may be useful.

- Please interpret the study based on the results presented in the abstract, emphasizing what is new without overstating your conclusions.

- Please avoid vague statements such as ""these results have major implications for policy/clinical care"". Mention only specific implications substantiated by the results.

- Please avoid assertions of primacy (""We report for the first time...."")"

AUTHOR SUMMARY

* It seems that the Author Summary of the previous version was not incorporated into the main text. Please revise. The Author Summary should follow the Abstract.

* In the author summary, please revise formatting and ensure you use bullet points.

METHODS AND RESULTS 

* Please consider reducing the number of subheadings. 

* Ethics: Please clarify whether the need for ethical approval was waived for your study and if so, why. Please provide the name(s) of the institutional review board(s) together with the approval numbers that provided ethical approval.

* “The KLG assessments for radiographs were performed by Dr. Piran Aliabadi, MD and Dr. Burt Sack, MD, under the direction of Dr. David Felson, MD from the Boston University Clinical Epidemiology Research and Training Unit for the baseline through 24-month visits.” – We think it would be useful to add descriptions of job level and experience.

* “Seventeen types of serum and urine biochemical markers were included in the FNIH cohort study, as detailed in reference (12).” – we suggest describing these in detail here instead of using a reference only. 

* l.320: Please clarify whether the clinical variable was ‘race’, ‘ethnicity’ or ‘race/ethnicity’.

* “At the baseline visit, 178 (61%) females were included in the development cohort 1, while 171 (57%) females were included in the test cohort 1 (Table S2).” – The description you have provided seems misleading. Please revise. Suggestion: “At the baseline visit, 293 (178/293, 61% females) individuals were included in the development cohort 1, while 301 (171/301, 57% females) individuals were included in the test cohort 1 (Table S2).”

* The terms gender and sex are not interchangeable (as discussed in https://www.who.int/health-topics/gender#tab=tab_1 ); please use the appropriate term.

* l.493: “This analysis will identify the most influential parameters and quantify uncertainty, improving the model's reliability and understanding its behavior in predicting KOA progression.” – The use of future tense seems wrong – please revise.

* Please check that any use of statistical terms (such as trend or significant) are supported by the data, and if not please remove them. 

* Figure 1: Please consider splitting the figure into two separate figures. 

* Figure 3: “The colored dotted line of yellow, orange, purple, green, blue, teal, and red represented the predictive performance change of Liu, Zhao, Cao, J Li, Chen, X Wang, Dang, and M Zhang, respectively.” – is it necessary to be able to identify the performer? We suggest removing the names and simply stating that each color belongs to one of the test individuals.

* Figure 3: For H and I, why is there no heading (total test cohort)? 

DISCUSSION

* Pleas remove all subheadings.

* We don’t think that it’s necessary to cite the ORs of other studies in parentheses throughout the discussion. 

“Our predictive model shows clinical potential in managing KOA progression by integrating MRI radiomic features, biomarkers, and clinical data.” – Based on the discussion and its limitations, do you think this statement is supported, or would you rather say that the model is a step toward developing a model that could be used in a clinic? We feel it would be better to tone down any statement about clinical utility.

General Editorial Requests

---

## [Editor Report · Decision Letter 4]

Dear Dr Li, 

On behalf of my colleagues and the Academic Editor, Christelle Nguyen, I am pleased to inform you that we have agreed to publish your manuscript "Predicting KOA Progression using Neural network withLongitudinal MRI Radiomics, and Biochemical Biomarkers: A Modeling Study" (PMEDICINE-D-24-02732R4) in PLOS Medicine.

I appreciate your thorough responses to the reviewers' and editors' comments throughout the editorial process. We look forward to publishing your manuscript. Editorially, there are a few remaining points that should be addressed prior to publication. We will carefully check whether the changes have been made. If you have any questions or concerns regarding these final requests, please feel free to contact me at atosun@plos.org.

Please see below the minor points that we request you respond to:

1) Title: Please change 'KOA' to 'knee osteoarthritis'.

2) Abstract: Please remove claims of primacy, such as 'novel'.

3) Author Summary: In the final bullet point of 'What Do These Findings Mean?', please state the main limitations of the study in non-technical language.

4) Funding statement/Financial Disclosure Statement: Please include URLs to all sponsors’ websites in the statement in the online submission form.

5) Ethics Statement: Please update the statement in the online submission form using the details provided in lines 68-75 of the last track changes version and in your last rebuttal. This is particularly important for the information regarding consent and the fact that no additional ethical approval was required for this study.

Before your manuscript can be formally accepted you will need to complete some formatting changes, which you will receive in a follow up email (including the editorial points above). Please be aware that it may take several days for you to receive this email; during this time no action is required by you. Once you have received these formatting requests, please note that your manuscript will not be scheduled for publication until you have made the required changes.

PRESS

Sincerely, 

Alexandra Tosun, PhD 

Senior Editor 

PLOS Medicine